# No Reason for No Supervision: Improved Generalization in Supervised Models

**Mert Bulent Sariyildiz**[1,2]    **Yannis Kalantidis**[1]    **Karteek Alahari**[2]    **Diane Larlus**[1]

[1] NAVER LABS Europe        [2] Univ. Grenoble Alpes, Inria, CNRS, Grenoble INP, LJK

## Abstract

We consider the problem of training a deep neural network on a given classification task, e.g., ImageNet-1K (IN1K), so that it excels at both the training task as well as at other (future) transfer tasks. These two seemingly contradictory properties impose a trade-off between improving the model's generalization and maintaining its performance on the original task. Models trained with self-supervised learning tend to generalize better than their supervised counterparts for transfer learning; yet, they still lag behind supervised models on IN1K. In this paper, we propose a supervised learning setup that leverages the best of both worlds. We extensively analyze supervised training using multi-scale crops for data augmentation and an expendable projector head, and reveal that the design of the projector allows us to control the trade-off between performance on the training task and transferability. We further replace the last layer of class weights with class *prototypes* computed on the fly using a memory bank and derive two models: **t-ReX** that achieves a new state of the art for transfer learning and outperforms top methods such as DINO and PAWS on IN1K, and **t-ReX\*** that matches the highly optimized RSB-A1 model on IN1K while performing better on transfer tasks.

Code and pretrained models: https://europe.naverlabs.com/t-rex

## 1 Introduction

Deep convolutional neural networks trained on large annotated image sets like ImageNet-1K (IN1K) (Russakovsky et al., 2015) have shown strong generalization properties. This motivated their application to a broad range of transfer tasks including the recognition of concepts that are not encountered during training (Donahue et al., 2014; Razavian et al., 2014).

Recently, models trained in a self-supervised learning (SSL) framework have become popular due to their ability to learn without manual annotations, as well as their capacity to surpass supervised models in the context of transferable visual representations. SSL models like MoCo (He et al., 2020), SwAV (Caron et al., 2020), BYOL (Grill et al., 2020) or DINO (Caron et al., 2021) exhibit stronger transfer learning performance than models (Wightman et al., 2021) trained on the same data with annotations (Sariyildiz et al., 2021).

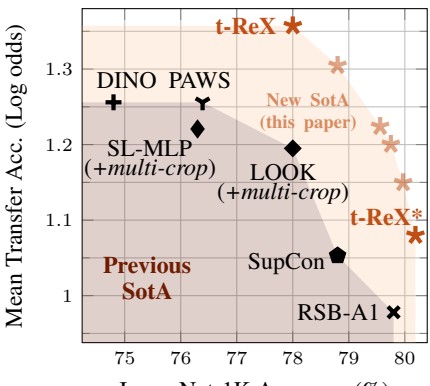

Figure 1: We present **t-ReX** and **t-ReX\***, two ResNet50 models trained with an improved supervised learning setup on ImageNet (IN1K), with strong performance on both transfer learning (y-axis, averaged over 13 tasks) and IN1K (x-axis).

This achievement is on the one hand exciting, as SSL approaches do not require an expensive and error-prone annotation process, but also seemingly counter-intuitive (Wang et al., 2022b) as it suggests that access to additional information, i.e., image labels, actually hinders the generalization properties of a model. Models learned via SSL are however not able to match their supervised counterparts on IN1K classification, i.e., on the concepts seen during training. Top-performing SSL and semi-supervised methods like DINO (Caron et al., 2021) or PAWS (Assran et al., 2021) still result in 3-5% lower top-1 accuracy compared to optimized supervised models such as RSB-A1 (Wightman et al., 2021).

In this paper, we argue that access to more information (in the form of manual annotations) should not hurt generalization, and we seek to improve the transferability of encoders learned in a supervised manner, while retaining their state-of-the-art performance on the supervised training task. The mismatch observed between IN1K and transfer performance suggests that this goal is not trivial. It has been shown, for example, that popular regularization techniques such as Label Smoothing (Szegedy et al., 2016), Dropout (Srivastava et al., 2014) or CutMix (Yun et al., 2019), which improve IN1K performance, actually lead to less transferable representations (Kornblith et al., 2021; Sariyildiz et al., 2021), and that representations learned on top of models underfitting their original task transfer better (Zhang et al., 2022).

We identify two key training components from the most successful SSL approaches that may lead to more transferable representations: multi-crop data augmentation (Caron et al., 2020) and the use of an expendable projector head, i.e., an auxiliary module added after the encoder during training and discarded at test time (Chen et al., 2020a). We study the impact of these two components on the transfer performance together with the performance on the training task, and present novel insights on the role of the projector design in this context. Furthermore, inspired by recent work on supervised learning (Feng et al., 2022; Khosla et al., 2020), we introduce *Online Class Means*, a memory-efficient variant of the Nearest Class Means classifier (Mensink et al., 2012) that computes class prototypes in an "online" manner with the help of a memory queue. This further increases performance. We perform an extensive analysis on how each component affects the learned representations, and look at feature sparsity and redundancy as well as intra-class distance. We also study the training dynamics and show that class prototypes and classifier weights change in different ways across iterations.

We single out the two ResNet50 instantiations that perform best at one of the two dimensions (transfer learning and IN1K), denoted as **t-ReX** and **t-ReX\***. **t-ReX** exceeds the state-of-the-art transfer learning performance of DINO (Caron et al., 2021) or PAWS (Assran et al., 2021) and still performs much better than these two on IN1K classification. **t-ReX\*** outperforms the state-of-the-art results of RSB-A1 (Wightman et al., 2021) on IN1K while generalizing better to transfer tasks. We visualize the performance of these two selected models, together with those of other top-performing configurations from our setup in Fig. 1, and compare it to state-of-the-art supervised, semi-supervised and self-supervised learning methods, across two dimensions: IN1K accuracy and mean transfer accuracy across 13 transfer tasks. This intuitively conveys how the proposed training setup *pushes the envelope* of the training-versus-transfer performance trade-off (from the "Previous SotA" region, to the "New SotA" one in Fig. 1) and offers strong pretrained visual encoders that future approaches could build on.

**Contributions.** We propose a supervised training setup that incorporates multi-crop data augmentation and an expendable projector and can produce models with favorable performance both on the training task of IN1K and on diverse transfer tasks. We thoroughly ablate this setup and reveal that the design of the projector allows to control the performance trade-off between these two dimensions, while a number of analyses of the features and class weights give insights on how each component of our setup affects the training and learned representations. We also introduce *Online Class Means*, a prototype-based training objective that increases performance even further and gives state-of-the-art models for transfer learning (**t-ReX**) and IN1K (**t-ReX\***).

## 2 RELATED WORK

Visual representations learned by deep networks for IN1K classification can transfer to other tasks and datasets (Donahue et al., 2014; Razavian et al., 2014). This generalization capability of networks has motivated researchers to propose practical approaches for measuring transfer learning (Goyal et al., 2019; Pándy et al., 2022; Zhai et al., 2019) or contribute to a formal understanding of generalization properties (Kornblith et al., 2019; Tripuraneni et al., 2020; Yosinski et al., 2014). Recent work in this context (Kornblith et al., 2021; Sariyildiz et al., 2021) shows that the best representations for IN1K are not necessarily the ones transferring best. For instance, some regularization techniques or loss functions improving IN1K classification lead to underwhelming transfer results. A parallel line of work based on self-supervised learning (Caron et al., 2020; Chen et al., 2020a; Grill et al., 2020) focuses on training models without manual labels, and demonstrates their strong generalization capabilities to many transfer datasets, clearly surpassing their supervised counterparts (Sariyildiz et al., 2021). Yet, as expected, SSL models are no match to the supervised models on the IN1K classification task itself.

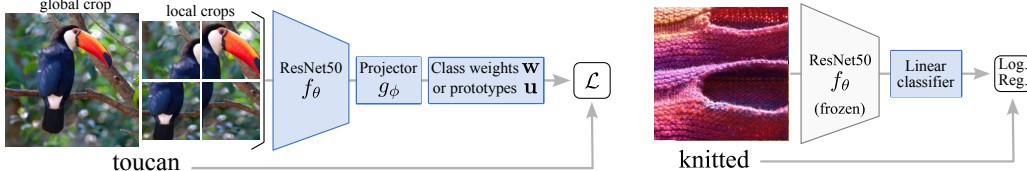

(a) Supervised learning using multi-crop and a projector.     (b) Transfer learning with a frozen model.

Figure 2: **Our proposed supervised learning setup** borrows multi-crop (Caron et al., 2020) and projectors (Chen et al., 2020a) from SSL to train on IN1K (*left*). The projector $g$ is discarded after training, and the ResNet backbone $f$ is used as a feature extractor in combination with a linear classifier trained for each task, e.g., for texture classification on DTD (Cimpoi et al., 2014) (*right*).

A few approaches tackle the task of training supervised models that also transfer well and share motivation with our work. SupCon (Khosla et al., 2020) extends SimCLR (Chen et al., 2020a) using image labels to build positive pairs. As such, its formulation is close to neighborhood component analysis (NCA) (Goldberger et al., 2004). It circumvents the need for large batches by adding a momentum and a memory similar to MoCo (He et al., 2020). Supervised-MoCo (Zhao et al., 2021) filters out false negatives in the memory bank of MoCo using image labels, while LOOK (Feng et al., 2022) modifies the NCA objective to only consider the closest neighbors of each query image. We experimentally observe that our model design leads to better transfer than all these works.

In this work, we propose an effective training setup, which leverages multi-crop augmentation (Caron et al., 2020) and an expendable projector head (Chen et al., 2020a), two key components in many successful SSL approaches. Creating multiple augmented versions (a.k.a. crops) of images in a batch was first proposed by Hoffer et al. (2020). Caron et al. (2020) further consider crops with different scales and resolutions in a self-supervised learning setting, creating challenging views of an image for which the model is encouraged to learn consistent representations (Assran et al., 2021; Caron et al., 2021). Recent work argues that multi-crop increases representation variance, is useful for online self-distillation (Wang et al., 2022a), and improves vision and language pretraining (Ko & Gu, 2022). We show that multi-crop over different resolutions works out-of-the-box also for supervised training on IN1K.

Using features from intermediate layers of networks has been considered before, e.g., for training object detectors (Lin et al., 2017) and image classification models (Lee et al., 2015), or evaluating the transferability of individual layers (Zhang et al., 2016; Gidaris et al., 2018) or groups of layers (Evci et al., 2022). However, selecting optimal layers for each problem is infeasible due to the computational nature of this selection. SimCLR (Chen et al., 2020a) proposed instead to rely on an expendable projector, a design that is now common practice in SSL (Zhou et al., 2022; Zbontar et al., 2021), and is starting to be adopted by supervised approaches like SupCon (Khosla et al., 2020) and LOOK (Feng et al., 2022). The impact of these projectors on the representation quality has only seldomly been studied. Wang et al. (2022b) have looked at the impact of projectors, but only for transfer and in isolation. Our work goes one step further and studies how projectors affect performance both on the *training task* and for transfer. We ablate many projector designs and study them jointly with multi-crop. Through our study, we uncover how useful projectors are at navigating the trade-off between training and transfer performance, leading to state-of-the-art results on both dimensions.

## 3   AN IMPROVED TRAINING SETUP FOR SUPERVISED LEARNING

We now present an improved training setup for learning supervised models that achieve high performance on *both* IN1K classification and a diverse set of transfer tasks.

Our setup trains a model (or *encoder*) $f_\theta$, parameterized by $\theta$. This model encodes an image $\mathbf{I}$ into a transferable representation $\mathbf{x} \in \mathbb{R}^d$. We follow the common protocol (Feng et al., 2022; Kornblith et al., 2021) and train all variants of our model on IN1K using a ResNet50 (He et al., 2016) encoder. This choice of encoder is influenced by recent observations (Wightman et al., 2021) that carefully optimized ResNet50 models perform on par with the best Vision Transformers (ViTs, Beyer et al. (2022)) of comparable size on IN1K. After training our models, we perform transfer learning. We

freeze the model's parameters so they are only used to produce transferable representations $(\mathbf{x})$, to be appended with a linear classifier for each transfer task (e.g., IN1K or any other dataset, see Fig. 2b).

Our improved training setup enriches the standard supervised learning paradigm with multi-crop augmentation and an expendable projector head (see Fig. 2a). We train our models with one of the two following training objectives: the standard softmax cross entropy loss that learns class weights, or an online variant of nearest class means that is based on class prototypes computed on-the-fly from a memory bank. We detail all the proposed improvements below.

**Multi-crop data augmentation.** Caron et al. (2020) leveraged many image crops of multiple scales and different resolutions when learning invariance to data augmentation in the context of SSL. Their data augmentation pipeline, termed *multi-crop*, is defined over two sets of *global* and *local* crops that respectively retain larger and smaller portions of an image. These crops are processed at different resolutions. We adapt this component to our supervised setup. Given an input image $\mathbf{I}$, we define two scale parameters, for global and local crops, which determine the size ratio between random crops and the image $\mathbf{I}$. We follow Caron et al. (2021) and resize global and local crops to $224 \times 224$ and $96 \times 96$, respectively. We extract multiple global and local crops, respectively $M_g$ and $M_l$. Fig. 2a illustrates one global $M_g = 1$ and four local $M_l = 4$ crops. In Sec. 4, we explore the use of multi-crop for supervised learning, and study the effect of different hyper-parameters under that setting.

**Expendable projector head.** To countervail the lack of annotations, SSL approaches tackle proxy tasks, such as learning augmentation invariance. In order to prevent the encoder from learning representations that overfit to a potentially unimportant pretext task, SSL architectures often introduce an expendable projector between the encoder and the loss function. On the contrary, for supervised learning, performance on the training task is a major goal in its own right. Here, we aim to learn supervised models that perform well on the training *and* on transfer tasks. These two requirements are not aligned and it is necessary to find a trade-off (Kornblith et al., 2021).

We argue that one can control this trade-off using an additional projector in the context of supervised learning. Similar to SSL methods (Chen et al., 2020a;b; Chen & He, 2021) and to the recent SL-MLP (Wang et al., 2022b) we introduce a Multi Layer Perceptron (MLP) projector as part of our supervised training pipeline. Let $g_\phi : \mathbb{R}^d \to \mathbb{R}^{d_b}$ denote this projector, parameterized by $\phi$. $g_\phi$ is composed of an MLP with $L$ hidden layers of $d_h$ dimensions followed by a linear projection to a bottleneck of $d_b$ dimensions. Each hidden layer is composed of a sequence of a linear fully-connected layer, batch-normalization (Ioffe & Szegedy, 2015) and a GeLU (Hendrycks & Gimpel, 2016) non-linearity. We further apply $\ell_2$-normalization to the output of $g_\phi$ and optionally also to the input. We illustrate this architecture in Fig. 3. Note that SL-MLP (Wang et al., 2022b) uses a similar head but with only one hidden layer and no input or output $\ell_2$-normalization, so SL-MLP can be seen as a special case of our projector architecture. We compare to their design in Sec. 4 and investigate how the number and dimension of hidden layers among other

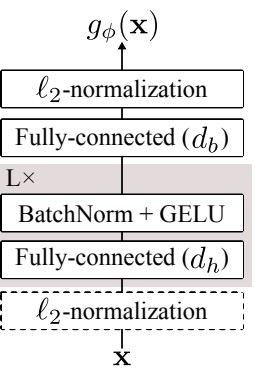

Figure 3: Architecture of the projector $g_\phi$.

design choices affect the transfer performance of the learned models, verifying and extending the findings of Wang et al. (2022b). On top of this, we study transfer performance in juxtaposition to performance on the *training task*, and derive the novel insight that projector design allows to control the trade-off between performance on the training task and transferability.

**Cosine softmax cross-entropy loss.** Incorporating both the components described above in a standard supervised learning paradigm, we can train with the standard softmax cross entropy loss using class labels. The training pipeline is illustrated in Fig. 2a. It uses multi-crop data augmentation on each input image $\mathbf{I}$ to produce $M = M_g + M_l$ crops $\mathbf{I}_j$, $j = 1, \ldots, M$. Each crop is individually input to the network composed of the encoder followed by the projector, and produces an embedding $\mathbf{z_j} = g_\phi(f_\theta(\mathbf{I}_j))$. To predict class labels, we multiply embeddings with trainable class weights $\mathbf{W} = \{\mathbf{w}_c \in \mathbb{R}^{d_b}\}_{c=1}^{C}$, where $C$ is the number of classes. We train the whole pipeline using the *cosine softmax* loss as it was shown to improve IN1K performance (Kornblith et al., 2021):

$$\mathcal{L}_{\text{CE}} = -\frac{1}{M} \sum_{j=1}^{M} \sum_{c=1}^{C} \mathbf{y}_{[c]} \log \frac{\exp(\mathbf{z_j}^\top \bar{\mathbf{w}}_c / \tau)}{\sum_{k=1}^{C} \exp(\mathbf{z_j}^\top \bar{\mathbf{w}}_k / \tau)}, \quad (1)$$

where $\mathbf{y} \in \{0,1\}^C$ is the $C$-dim one-hot label vector corresponding to image $\mathbf{I}$, $\tau$ is a temperature hyper-parameter and $\bar{\mathbf{w}}_c = \mathbf{w}_c / \|\mathbf{w}_c\|$. Note that projector outputs $\mathbf{z}$ are already $\ell_2$-normalized.

**Online Class Means.** Motivated by the recent success of momentum encoders as a way of maintaining online memory banks for large-scale training (He et al., 2020), we revisit the prototype-based Nearest Class Means (NCM) approach of Mensink et al. (2012) and introduce a memory-efficient variant that computes class prototypes in an "online" manner with the help of a memory queue.

Concretely, following Mensink et al. (2012), we define $\mathbf{u}_c$ to be the class prototype or *class mean* for class $c$, i.e., the mean of all embeddings from that class, and define $\mathbf{U} = \{\mathbf{u}_c\}_{c=1}^C$. Given that we jointly learn class means and the embeddings, computing the exact mean at each iteration is computationally prohibitive. Instead, we formulate an *online* version of NCM that uses a memory bank $\mathcal{Q}$ which stores $\ell_2$-normalized embeddings $\mathbf{z}$ output by the projector, similar to the memory bank from MoCo (He et al., 2020). Given the memory $\mathcal{Q}$, we do not learn class weights, but instead compute a *prototype* for each class, on-the-fly, as the average of the embeddings in the memory which belong to that class. Formally, if $\mathcal{Q}_c$ denotes samples in memory that belong to class $c$, and $N_c = |\mathcal{Q}_c|$, the loss function becomes:

$$\mathcal{L}_{\text{OCM}} = -\frac{1}{M} \sum_{j=1}^M \sum_{c=1}^C \mathbf{y}_{[c]} \log \frac{\exp(\mathbf{z_j}^\top \bar{\mathbf{u}}_c / \tau)}{\sum_{k=1}^C \exp(\mathbf{z_j}^\top \bar{\mathbf{u}}_k / \tau)}, \text{ with } \bar{\mathbf{u}}_c = \frac{\mathbf{u}_c}{\|\mathbf{u}_c\|} \text{ and } \mathbf{u}_c = \frac{1}{N_c} \sum_{\mathbf{z} \in \mathcal{Q}_c} \mathbf{z}. \quad (2)$$

We refer to the above training objective as *Online Class Means* or *OCM*. To make sure the embeddings stored in the memory remain relevant as the encoder is updated during training, we follow MoCo (He et al., 2020) and store in memory embeddings from an exponential moving average (EMA) model trailing $f_\theta$ and $g_\phi$. As we show in our analysis in Sec. 4.2, estimating class prototypes using only the relatively small subset of samples in the memory bank leads to class prototypes that drift more across iterations compared to SGD-optimized class weights that converge faster. Further details as well as other variants are discussed in Appendix A.

## 4 EXPERIMENTS

In Sec. 4.1, we exhaustively study the design of the main components of our setup, i.e., multi-crop augmentation, projectors, and OCM. This leads to a summary of our main findings. We then analyze the learned representations, class weights, and prototypes in Sec. 4.2. There, we explore how each component affects several facets like feature sparsity and redundancy, average coding length, as well as intra-class distance. We also study training dynamics like gradient similarity for multi-crop or how prototypes and classifier weights change across iterations for OCM. Finally, in Sec. 4.3 we plot the performance of multiple variants of the proposed training setup on the training-versus-transfer performance plane, empirically verifying its superiority over the previous state of the art.

**Protocol.** All our models are trained on the training set of ImageNet-1K (IN1K) (Russakovsky et al., 2015). Due to the computational cost of training models on IN1K, each configuration is trained only once. Given an IN1K-trained model, we discard all the training-specific modules (e.g., the projector $g_\phi$, the class weights $\mathbf{W}$), and use the encoder $f_\theta$ as a feature extractor, similar to Kornblith et al. (2019); Sariyildiz et al. (2021). For each dataset we evaluate on, we learn a linear logistic regression classifier with the pre-extracted features and independently optimize each classifier's hyper-parameters *for every model and every evaluation dataset* using Scikit-learn (Pedregosa et al., 2011) and Optuna (Akiba et al., 2019) (see details in Appendix B). We repeat this process 5 times with different random seeds and report the average accuracy (variance is negligible). Note that the feature extractor is never fine-tuned, and, because we start from pre-extracted features, no additional data augmentation is used when learning the linear classifiers.[1] This protocol is illustrated in Fig. 2b.

**Evaluation datasets and measures.** We measure performance on the training task by evaluating classification accuracy on the IN1K validation set. To evaluate transfer learning, we measure classification performance on 13 datasets: the 5 ImageNet-CoG datasets (Sariyildiz et al., 2021)

---

[1]Although in their evaluation Caron et al. (2021); Zhai et al. (2019) train linear classifiers with data augmentation or fine-tune the encoder while training classifiers, we found that such protocols make a proper hyper-parameter validation computationally prohibitive. We instead follow the linear evaluation protocol from Kornblith et al. (2019) and Sariyildiz et al. (2021).

Figure 4: **Impact of the number of local crops** ($M_l$) on the performance on IN1K (*left*) and transfer datasets (*right*) when varying the number of hidden layers $L$ in the projector and $M_g$=1.

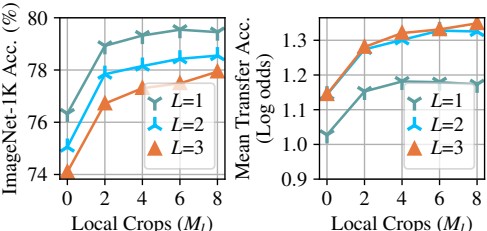

Table 1: **Impact of the projector size** on performance, via the number of hidden layers $L$ (*left*) and hidden units $d_h$ (*right*). The default configuration: $L$=1, $d_h$=2048, $d_b$=256 and with $\ell_2$-normalization of the input (highlighted rows). We use $M_g$=1 and $M_l$=8 ("Base+Mc").

| | IN1K | Transfer | $d_h$ | IN1K | Transfer |
|---|---|---|---|---|---|
| Base | 76.6 | 0.10 | 512 | 80.0 ↑ | 0.82 |
| Base+Mc | 79.7 | 0.25 | 1024 | 80.0 | 1.06 |
| $L=1$ | 79.8 ↑ | 1.15 | 2048 | 79.8 | 1.15 |
| $L=2$ | 78.6 | 1.31 | 4096 | 79.8 | 1.20 |
| $L=3$ | 77.5 | 1.33 ↓ | 8192 | 79.4 | 1.22 ↓ |

that measure concept generalization, and 8 commonly used smaller-scale datasets: Aircraft (Maji et al., 2013), Cars196 (Krause et al., 2013), DTD (Cimpoi et al., 2014), EuroSAT (Helber et al., 2019), Flowers (Nilsback & Zisserman, 2008), Pets (Parkhi et al., 2012), Food101 (Bossard et al., 2014) and SUN397 (Xiao et al., 2010). We report two metrics: Top-1 accuracy on IN1K and transfer accuracy via log-odds (Kornblith et al., 2019) averaged over the 13 transfer datasets. Note that we provide details on the datasets, the exact log-odds formulation, and per dataset results in the Appendix, respectively in Tab. 4, Appendix B.3, and Tab. 7. Appendices C.7 and C.8 present additional evaluations on IN1K-Sketch (Wang et al., 2019), IN1K-v2 (Recht et al., 2019) and two long-tail datasets: i-Naturalist 2018 and 2019 (Van Horn et al., 2018).

**Implementation details.** $f_\theta$ is a ResNet50 (He et al., 2016) encoder, trained for 100 epochs with mixed precision in PyTorch (Paszke et al., 2019) using 4 GPUs where batch norm layers are synchronized. We use an SGD optimizer with 0.9 momentum, a batch size of 256, 1e-4 weight decay and a learning rate of $0.1 \times$ batch size$/256$, which is linearly increased during the first 10 epochs and then decayed with a cosine schedule. We set $\tau = 0.1$ and, unless otherwise stated, we use the data augmentation pipeline from DINO (Caron et al., 2021) with 1 global and 8 local crops ($M_g = 1$ and $M_l = 8$). A detailed list of the training hyper-parameters is given in the Appendix (Tab. 3). Training one of our models takes up to 3 days with 4 V100 GPUs depending on its projector configuration.

## 4.1 ANALYSIS OF COMPONENT DESIGN AND HYPER-PARAMETERS

**Multi-crop data augmentation.** We first study the effect of the number of local crops on IN1K and transfer performance. We train supervised models using Eq. (1) with 1 global and 2, 4, 6 or 8 local crops, and projectors composed of 1, 2 or 3 hidden layers, and report results in Fig. 4. Our main observations are: a) training with local crops improves the performance on both IN1K and transfer tasks, and b) although increasing the number of local crops generally helps, performance saturates with 8 local crops. We set $M_g = 1$ and $M_l = 8$ for all subsequent evaluations. Further ablations using different scale and resolution parameters are presented in Appendix C.3.

Note that using local crops increases the effective batch size, which, in turn, increases training time. We therefore conduct two experiments to see if a longer training or a larger batch size would lead to similar gains. We train two models using a single crop sampled from a wide scale range (i.e., able to focus on both large and small image regions), one with 9×larger batch size, the other for 800 epochs. Unlike multi-crop, these models bring no significant gain (see Tab. 8 in the Appendix for details).

**Expendable projector head.** We study the impact of different architectural choices and hyper-parameters for the projector. We vary the number of hidden layers ($L$), the dimension of the hidden ($d_h$) and bottleneck ($d_b$) layers, and whether or not to $\ell_2$-normalize the projector input ($\ell_2$). We start from a default configuration: $L = 1$, $d_h = 2048$, $d_b = 256$ and with $\ell_2$-normalized inputs. We ablate each parameter separately by training models optimizing Eq. (1). We use multi-crop in all cases.

The most interesting results from this analysis are presented in Tab. 1. We see that *the number of hidden layers ($L$) is an important hyper-parameter that controls the trade-off between IN1K and transfer performance.* Adding a projector head with a single hidden layer not only improves the already strong IN1K performance of multi-crop (Base+Mc in Tab. 1), but also significantly boosts its average transfer performance. More hidden layers seem to increase transfer performance, at the cost of a decrease in IN1K accuracy. The same can be said about the dimension of the hidden

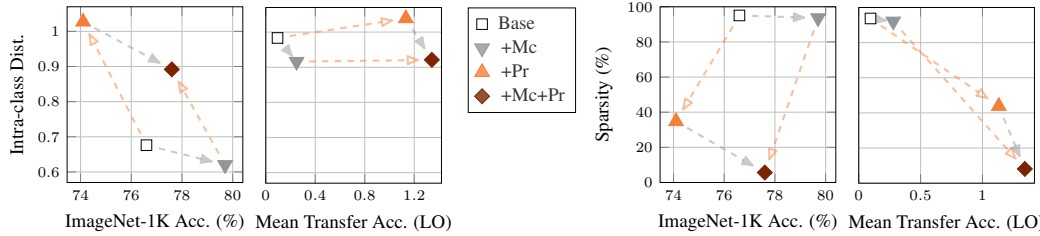

Figure 5: Average intra-class $\ell_2$-distance between samples from the same class (*left*) and sparsity as the percentage of feature dimensions close to zero (*right*), on IN1K and averaged over transfer datasets. Gray and Orange arrows denote changes due to adding multi-crop and projectors, respectively.

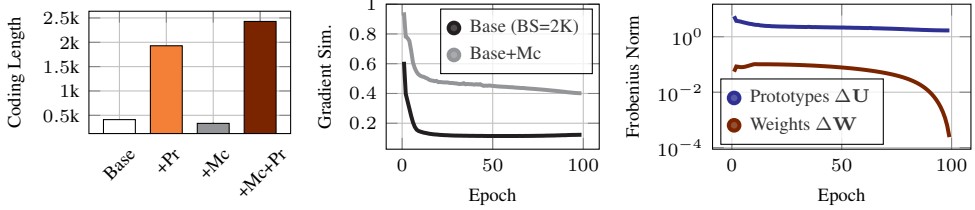

Figure 6: (*left*) Average coding length per sample (Yu et al., 2020) over all *transfer* datasets. (*middle*) Average similarity between class weight gradients $\nabla_{\mathbf{w}_c}\mathcal{L}_{\text{CE}}$ during training. (*right*) Change in class weights $\mathbf{W}$ and prototypes $\mathbf{U}$ at every iteration across all classes (see text for details) for models trained using Eq. (1) and Eq. (2), respectively.

layer, yet we further see that a larger $d_h$ significantly increases transfer performance, and moderately decreases IN1K accuracy. On the contrary, we observe that the bottleneck dimension $d_b$ and input $\ell_2$-normalization only have a small influence on IN1K and transfer performance (see Appendix C.4). Overall, our observations verify and significantly extend the ones recently presented by Wang et al. (2022b). We not only study the design of projectors jointly with multi-crop, but also analyse transfer performance jointly with performance on IN1K, revealing a *trade-off* between the two, that is fully controlled through the design of the project head.

**Online class means.** There are two main hyper-parameters in OCM: the size of the memory bank and the momentum of the EMA models that populate the memory bank and provide the embeddings for class prototypes. We explored momentum values 0 and 0.999 (in the former, we directly use $f_\theta$ and $g_\phi$ to compute prototypes) and see that trailing EMA is essential for maintaining high performance, i.e., momentum $= 0$ performs poorly, aligning with the observations for MoCo (He et al., 2020). However, unlike MoCo or other recent methods such as LOOK (Feng et al., 2022), OCM does not require a large memory bank to achieve the highest performance. We experimented with memory sizes between 2048 and 65546, and found that 8192 works best (see Fig. 12b in Appendix).

### 4.2 ANALYSIS OF LEARNED FEATURES, CLASS WEIGHTS AND PROTOTYPES

We now investigate how different components of our setup affect training or the learned representations. We analyse the features produced, class weights and prototypes from the following models: a) *Base*: a model trained using cosine softmax loss without multi-crop and projector, b) *Base (BS=2K)*: Base but with $9\times$larger batch size, c) *Base+Mc*: Base with multi-crop, d) *Base+Pr*: Base with a projector, e) *Base+Mc+Pr*, and f) *OCM*: a model trained using Eq. (2); details in Appendix C.6.

**Intra-class distance.** We start by analysing the $\ell_2$-normalized features for the four models, Base, Base+Mc, Base+Pr and Base+Mc+Pr, by computing the average $\ell_2$-distance between samples from the same class (i.e., intra-class distance). We see in Fig. 5 (left) that multi-crop reduces intra-class distance on IN1K, while projectors increase it. Not surprisingly, this correlates with training task performance, i.e., lower intra-class distance translates to better performance on IN1K. On the transfer datasets, however, we found no strong correlation between the two, i.e., transfer performance does not necessarily depend on intra-class distance.

**Sparsity.** In Fig. 5 (right) we report feature *sparsity ratio*, i.e., the percentage of feature dimensions close to zero for $\ell_2$-normalized features from the four models. We see that: a) the average sparsity

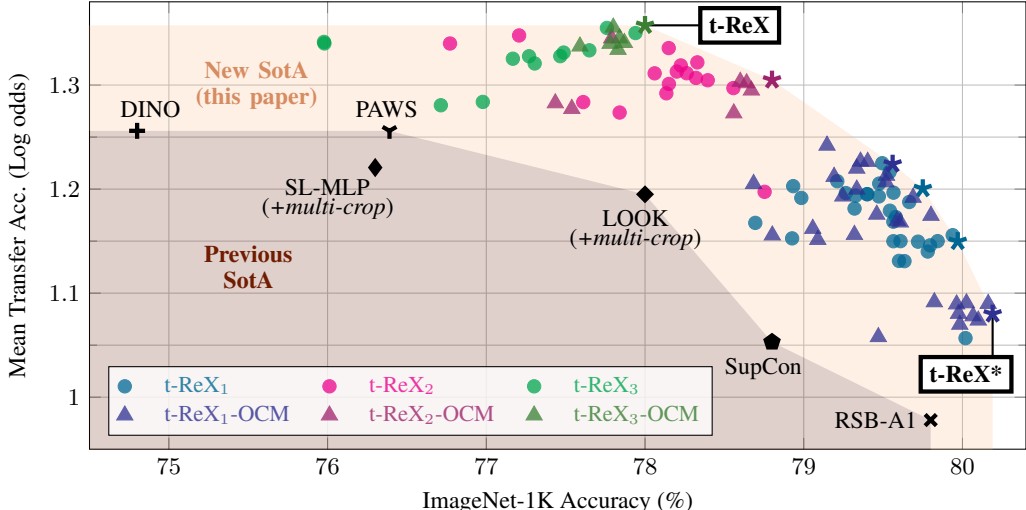

Figure 7: **Comparison on the training task vs transfer task performance for ResNet50.** We report IN1K (Top-1 accuracy) and transfer performance (log odds) averaged over 13 datasets (5 ImageNet-CoG levels, Aircraft, Cars196, DTD, EuroSAT, Flowers, Pets, Food101 and SUN397) for a large number of our models trained with the supervised training setup presented in Sec. 3. Models on the convex hull are denoted by stars. We compare to the following state-of-the-art (SotA) models: Supervised: RSB-A1, SupCon, SL-MLP and LOOK with multi-crop; self-supervised: DINO; semi-supervised: PAWS.

ratio on the transfer datasets is inversely correlated with performance, i.e., linear classifiers trained on *less sparse features achieve better transfer performance*, and b) *projectors dramatically reduce sparsity*. We find this last observation intuitive: features from the layer right before the cross-entropy loss are encouraged to be as close to a one-hot vector as possible and therefore sparse. Introducing projectors in between allows the encoder to output less sparse features, which improves transfer.

**Coding length.** To further investigate our observations on sparsity, we follow Yu et al. (2020) and compute the average coding length per sample on the transfer datasets (see Fig. 6 (left)). We see that projectors largely increase the "information content" of representations. This was also verified by analysing singular values per dimension for models with and without projectors. We observed that the feature variance is more uniformly distributed over dimensions when a projector is used (see Appendix C.6). These observations might explain why projectors reduce overfitting to IN1K concepts.

**Gradient similarity.** To understand why using multi-crop increases performance for the same batch size, we examine the gradients of class weights $\nabla_{\mathbf{W}} \mathcal{L}_{\mathrm{CE}}$ for two models that have the same effective batch size, with and without multi-crop. At each training iteration, we compute the average cosine similarity between individual gradients of every pair of class weights $\nabla_{\mathbf{W}_{c_i}} \mathcal{L}_{\mathrm{CE}}$ and $\nabla_{\mathbf{W}_{c_j}} \mathcal{L}_{\mathrm{CE}}$ for any $c_i \neq c_j$. As we see from Fig. 6 (middle), cosine similarity increases substantially with multi-crop. In other words, on average, classifier gradients (and therefore the class weights themselves) are more entangled. We attribute this to the fact that some of the local crops (e.g., the ones that mostly cover background and hence are not really discriminative for the class at hand) are harder to classify. This leads to a harder task and gradients of higher variance (shown also in Fig. 16 in the Appendix).

**Change in class weights and prototypes.** To understand the differences between the training objectives in Eq. (1) and Eq. (2), we measure how much class weights $\mathbf{W}$ and prototypes $\mathbf{U}$ change during the training phase. In Fig. 6 (right), we plot the average change over all classes by computing the Frobenius norm between before and after each iteration, i.e., $\Delta \mathbf{W} = \|\bar{\mathbf{W}}^t - \bar{\mathbf{W}}^{t-1}\|_2$ and $\Delta \mathbf{U} = \|\bar{\mathbf{U}}^t - \bar{\mathbf{U}}^{t-1}\|_2$, where $t$ is the training iteration, and $\bar{\mathbf{W}}$ and $\bar{\mathbf{U}}$ are the class weight $\mathbf{w}_c$ and prototype $\mathbf{u}_c$ $\ell_2$-normalized per class and concatenated, respectively. Interestingly, we observe that *prototypes $\mathbf{U}$ change orders of magnitude more than class weights $\mathbf{W}$ throughout training*. We believe this is because we compute class prototypes using only the small subset of images from our memory bank. The average number of samples per class on IN1K is 1281, whereas, on average we have only 8 per class in the memory bank. We argue that this prevents OCM from overfitting, leading to higher IN1K performance, as we show next.

### 4.3 PUSHING THE ENVELOPE OF TRAINING-VERSUS-TRANSFER PERFORMANCE

In this section we report and analyse results from more than 100 different models trained on IN1K, all different instantiations of the proposed training setup. We varied hyper-parameters such as the number of hidden layers in the expandable projector head or the training objective. The most important results are depicted in Fig. 7, while an extended version of this figure is presented in Fig. 10 in the Appendix.

**Previous state of the art.** RSB-A1 (Wightman et al., 2021) is a highly optimized supervised ResNet50 model with top performance on IN1K. The self-supervised DINO model (Caron et al., 2021) has shown top transfer learning performance, while also performing well on IN1K. The semi-supervised PAWS (Assran et al., 2021) model matches DINO in transfer performance, with improved IN1K accuracy. To our knowledge, RSB-A1 and DINO/PAWS are the current state-of-the-art ResNet50 models for IN1K classification and transfer learning respectively. We also compare to three recent supervised models: SupCon (Khosla et al., 2020), LOOK (Feng et al., 2022) and SL-MLP (Wang et al., 2022a). For all models except LOOK and SL-MLP, we evaluate the models provided by the authors. Due to the absence of official code we reproduced LOOK and SL-MLP ourselves, enhancing them with multi-crop. Our reproductions achieve higher performance than the one reported in the original papers. In both cases, we use a projector with 1 hidden layer.

**Notations.** Models trained with the basic version of the proposed training setup, i.e., using multi-crop, a projector with $L$ hidden layers and a cosine softmax cross-entropy loss are reported as **t-ReX**$_L$. For models using the OCM training objective we append **-OCM**. Models on the "envelope" (i.e., the convex hull) of Fig. 7 are highlighted with a star (exact configurations are in the Appendix: Tabs. 3 and 7).

**Main results.** Our main observations from results presented in Fig. 7 can be summarized as follows.

- *Pushing the envelope.* Many variants from our supervised training setup "push" the envelope beyond the previous state of the art, across both axes. Several of these models improve over the state of the art on one or the other axis, but no single model outperforms all the others on both dimensions. As the number of hidden layers of the projector increases, models gradually move from the lower right to the upper left corner of the plane. This shows again that increasing the projector complexity improves transfer performance at the cost of IN1K (training task) performance.
- *No reason for no supervision.* A large number of supervised variants outperform the DINO method with respect to transfer learning, while also being significantly better on IN1K. We therefore show that training with label supervision does not necessarily require to sacrifice transfer learning performance and one should use label information if available.
- *State-of-the-art IN1K performance with three simple modifications.* A number of t-ReX$_1$ models outperform the highly optimized RSB-A1 on IN1K, essentially by using only three components over the "vanilla" supervised learning process that is considered standard practice: a) cosine softmax with temperature, b) multi-crop data augmentation, and c) an expendable projector.
- *Training with class prototypes brings further gain.* Given the same projector configuration, training models with the OCM objective (Eq. (2)) has a small advantage over training with cosine softmax (Eq. (1)). We see that 4 of the 6 points on the convex hull in Fig. 7 are t-ReX-OCM models. This suggests that using class prototypes is a viable alternative to learning class weights end-to-end.
- *Introducing t-ReX and t-ReX\*.* We single out the two instantiations that respectively excel on the transfer learning and IN1K axes, i.e., **t-ReX**$_3$**-OCM** and **t-ReX**$_1$**-OCM**. We rename them **t-ReX** and **t-ReX\***, respectively. We envision these two **t**ransferable **Re**sNet50 models and their corresponding training setups to serve as strong supervised baselines for future research on transfer learning and IN1K. All the hyper-parameters for these two models are in Tab. 3 in the Appendix.

## 5 CONCLUSION

We present a supervised training setup that leverages components from self-supervised learning, and improves generalization without conceding on the performance of the original task, i.e., IN1K classification. We also show that substituting class weights with prototypes used an online class mean classifier over a small memory bank boosts performance even further. We extensively analyze the design choices and parameters of those models, and show that many variants push the envelope on the IN1K-transfer performance plane. This validates our intuition that image-level supervision, if available, can be beneficial to both IN1K classification and transfer tasks.

**Acknowledgements.** This work was supported in part by MIAI@Grenoble Alpes (ANR-19-P3IA-0003), and the ANR grant AVENUE (ANR-18-CE23-0011).

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

# APPENDIX

## CONTENTS

## A  FURTHER DETAILS ON THE IMPROVED SUPERVISED TRAINING SETUP

### A.1  DATA AUGMENTATION

Our training setup for improving the generalization performance of supervised models (described in Sec. 3 of the main paper) includes the multi-crop augmentation initially proposed in SwAV (Caron et al., 2020). In our experiments, we use the multi-crop implementation from DINO (Caron et al., 2021), which consists of three augmentation branches (two for global crops and one for local crops). The pipeline of this multi-crop augmentation is illustrated in Fig. 8a. We also illustrate the pipelines of the "vanilla" PyTorch and SimSiam augmentations that are used in some of our ablation studies in Fig. 8b and Fig. 8c respectively. For our experiments with two global crops, i.e., when $M_g = 2$ in Fig. 11, we use all three branches. In all the other experiments, which have only one global crop $M_g = 1$, we use two of the branches: the second global crop and the one for local crops. Tab. 2 summarizes the parameters of the augmentation operations used in our experiments.

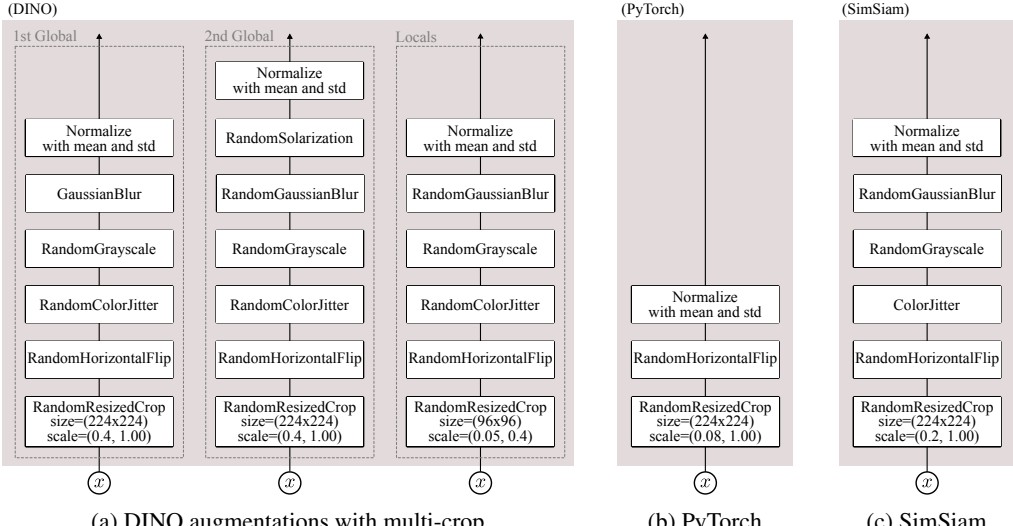

(a) DINO augmentations with multi-crop      (b) PyTorch      (c) SimSiam

Figure 8: **Data-augmentation pipelines** considered in our work. We use the multi-crop augmentation implemented in DINO (Caron et al., 2021) (a) as part of our improved training setup that is presented in Sec. 3 of the main paper. We also compare to the PyTorch (Marcel & Rodriguez, 2010; Paszke et al., 2019) (b) and SimSiam (Chen & He, 2021) (c) augmentations in our ablations, see Appendix C.3. The parameters of the operations in (a) are given in Tab. 2. We use the default values for the operations in (b) and (c). These pipelines are implemented using torchvision (Marcel & Rodriguez, 2010) and Python Imaging Library. Accordingly, the operation and parameter names follow the conventions from these open-source libraries.

Table 2: **Default parameters of the DINO augmentations** used in our experiments. We implement the RandomGaussianBlur and RandomSolarization using Python Imaging Library, and use the torchvision (Marcel & Rodriguez, 2010) implementations for the remaining ones. For RandomGaussianBlur, "radius" denotes a range from which we uniformly sample radius values. Note that some of these operations involve other parameters, and in these cases we use their default values. The scale parameter of RandomResizedCrop is different for **t-ReX**, see Tab. 3 for details.

| Augmentation Operation | Parameters |
|---|---|
| RandomResizedCrop for global crops | size=$(224 \times 224)$, scale=$(0.4, 1.0)$ |
| RandomResizedCrop for local crops | size=$(96 \times 96)$, scale=$(0.05, 0.4)$ |
| RandomHorizontalFlip | probability=0.5 |
| RandomColorJitter | probability=0.8, brightness=0.4, contrast=0.4, saturation=0.2, hue=0.1 |
| RandomGrayScale | probability=0.2 |
| RandomGaussianBlur | probability=0.2, radius=$(0.1, 2.0)$ |
| RandomSolarization | probability=0.2, threshold=128 |
| Normalization | mean=$(0.485, 0.456, 0.406)$, std=$(0.229, 0.224, 0.225)$ |

## A.2 ONLINE CLASS MEANS (OCM)

In Sec. 3 of the main paper, we introduce the "online" version of the Nearest Class Mean classifier (Mensink et al., 2012), referred to as Online Class Mean (OCM). In this section, we give a schematic illustration of this model and describe its implementation details.

**Illustrations of the different loss functions.** In Fig. 9 we visualize the supervised models. We train it using the two loss functions defined in Sec. 3 of the main paper. Fig. 9a and Fig. 9b correspond to the models for Eq. (1) and Eq. (2) of the main paper, respectively. As seen from Fig. 9b, and as explained in Sec. 3 of the main paper, OCM follows SupCon (Khosla et al., 2020) and LOOK (Feng et al., 2022) and uses the momentum network and memory queue proposed in He et al. (2020). We explain them in detail below.

**Momentum encoder $f_\xi$ and projector $g_\zeta$.** To keep a memory bank of slowly evolving features, we keep an exponential moving average (EMA) of the encoder $f_\theta$ and projector $g_\phi$ parameters. Concretely, momentum encoder $f_\xi$ and momentum projector $g_\zeta$, whose parameters are respectively

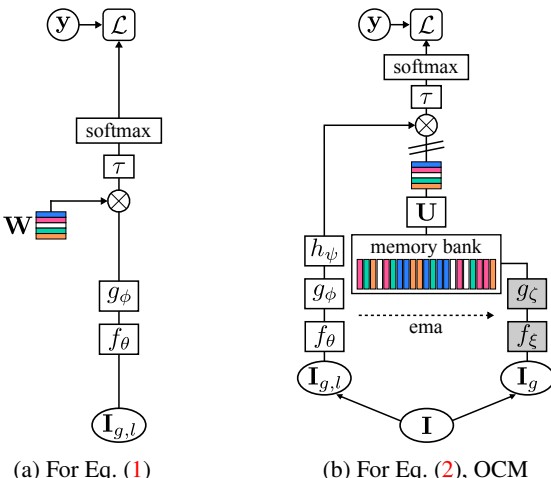

(a) For Eq. (1)        (b) For Eq. (2), OCM

Figure 9: **The supervised models** we train using our proposed setup. $\mathbf{I}_g$ and $\mathbf{I}_{g,l}$ represent only global crops or both global and local crops. Our **t-ReX** and **t-ReX\*** variants have the form shown in Fig. 9b, with projector configurations $L = 1$ (for **t-ReX\***) and $L = 3$ (for **t-ReX**), $d_h = 2048$, $d_b = 256$, input $\ell_2$-normalization enabled, memory bank size $\mathcal{Q} = 8192$ and with no predictor, i.e., $h_\psi$ is an identity mapping.

EMA of $\theta$ and $\phi$, are defined as: $\xi \leftarrow m \times \xi + (1 - m) \times \theta$ and $\zeta \leftarrow m \times \zeta + (1 - m) \times \phi$, where $m = 0.999$ is the momentum parameter. As shown in Fig. 9b, we only feed global crops $\mathbf{I}_g$ through these momentum networks $f_\xi$ and $g_\zeta$ during training. Both global and local crops $\mathbf{I}_{g,l}$ are passed through the encoder $f_\theta$ and projector $g_\phi$.

**Expendable predictor $h_\psi$ after projector $g_\phi$.** In several SSL methods with dual-network architectures, e.g., BYOL (Grill et al., 2020) and SimSiam (Chen & He, 2021), breaking the architectural symmetry, by adding a multi-layer perceptron to one of the branches, was shown to improve the generalization of representations. Following this practice, we also experimented with training OCM models optionally using an expendable predictor head $h_\psi : \mathbb{R}^{d_b} \to \mathbb{R}^{d_b}$ with parameters $\psi$, added after the projector head $g_\phi$ (as shown in Fig. 9b). These predictor heads contain fully-connected, batch-normalization (Ioffe & Szegedy, 2015) and GeLU (Hendrycks & Gimpel, 2016) layers, followed by another fully-connected layer and $\ell_2$-normalization. The first (resp. second) fully-connected layer maps from $\mathbb{R}^{d_b}$ to $\mathbb{R}^{d_p}$ (resp. $\mathbb{R}^{d_p}$ to $\mathbb{R}^{d_b}$). In our experiments $d_p$ is generally 2048 dimensions. See Appendix C.5 for a discussion on the impact of predictors on performance; Note that neither **t-ReX** nor **t-ReX\*** is trained with an expendable predictor head $h_\psi$.

**Memory bank.** The original NCM (Mensink et al., 2012) formulation require access to the entire dataset at each SGD training iteration, which is not possible in our case. To circumvent this, we use a memory queue $\mathcal{Q}$ which stores $\ell_2$-normalized momentum projector outputs $\mathcal{Q} = \{g_\zeta(f_\xi(\mathbf{I}_g))\}$ for global crops. In OCM, we compute a "prototype" for each class $c$, as the mean over all memory points from that class $\mathbf{u}_c = 1/N_c \sum_{\mathbf{z} \in \mathcal{Q}_c} \mathbf{z}$, where $\mathcal{Q}_c$ denotes samples in the queue that belong to class $c$ and $N_c = |\mathcal{Q}_c|$. Then, for a given training crop $\mathbf{I}_j$ (can be either a global or local crop), we compute its predictor outputs $h_\psi(g_\phi(f_\theta(\mathbf{I}_j)))$ and obtain class prediction scores by taking the inner product between this predictor output and the set of all class prototypes $\mathbf{U} = \{\mathbf{u}_c\}_{c=1}^{1000}$ as defined in Eq. (2) of the main paper. In Appendix C.5 we study the impact that the size of the memory bank has on IN1K and transfer performance.

**Class prototype neighbors.** In Tab. 12 we provide a visualization of the top-5 nearest prototypes for a random set of classes for **t-ReX\***.

## A.3   VARIANT: RANDOM ORTHOGONAL CLASSIFIERS (T-REX$_L$-*orth*)

In Sec. 3 of the main paper, we propose the OCM variant where class weights $\mathbf{W}$ are replaced by class means $\mathbf{U} = \{\mathbf{u}_c\}_{c=1}^C$ which are obtained in an online manner using a memory bank (see also Appendix A.2 for details). Inspired by Hoffer et al. (2018); Sariyildiz et al. (2020), we

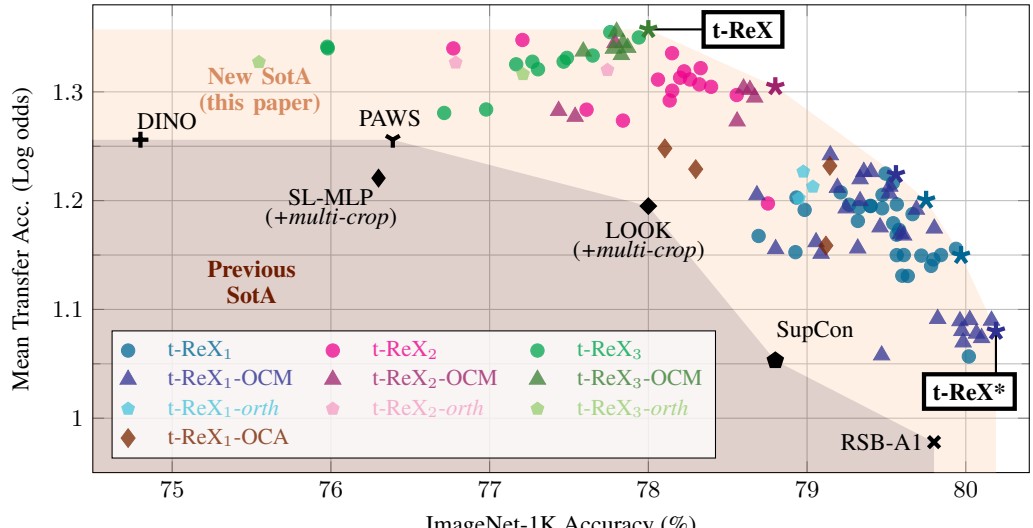

Figure 10: **Extended version of Fig. 7 of the main paper**, featuring the t-ReX-OCA and t-ReX$_L$-*orth* variants. We additionally plot six t-ReX$_L$-*orth* variants with and without projector input $\ell_2$-norm and $L = [1, 2, 3]$, and four t-ReX-OCA variants with different temperature parameter $\tau$ values and $L = 1$. We report IN1K (top-1 accuracy) and transfer performance (log odds) averaged over 13 datasets (see Tab. 4 for the list) for a large number of our models trained with the supervised training setup presented in Sec. 3 of the main paper. Models on the convex hull are denoted by stars. We compare to the following state-of-the-art (SotA) models: Supervised RSB-A1 (Wightman et al., 2021), SupCon (Khosla et al., 2020), and SL-MLP (Wang et al., 2022a) and LOOK (Feng et al., 2022) with multi-crop; Self-supervised DINO (Caron et al., 2021); Semi-supervised PAWS (Assran et al., 2021).

explore another variant of our setup where class weights $\mathbf{W} = \{\mathbf{w}_c\}_{c=1}^{1000}$ are initialized with random *orthogonal* vectors and kept frozen while the encoder $f_\theta$ and projector $g_\phi$ networks are trained. Our motivation is that such vectors may lead to higher class separation obtaining higher accuracy on the training task, as also noted by Kornblith et al. (2021).

To this end, without using momentum networks, a predictor head or a memory bank, we simply optimize Eq. (1) in the main paper with "fixed" class weights, and denote these models with the t-ReX$_L$-*orth* notation. Due to our limited computational budget, we train 6 t-ReX$_L$-*orth* variants with and without projector input $\ell_2$-norm, $L = [1, 2, 3]$, $d_h = 2048$ and $d_b = 256$, and plot their IN1K and average transfer accuracies along with the others from the main paper in Fig. 10. We observe that none of those six models were able to expand the envelope of training-versus-transfer performance.

## A.4   VARIANT: ONLINE COMPONENT ANALYSIS (T-REX-OCA)

Recent supervised learning methods like SupCon (Khosla et al., 2020) or LOOK (Feng et al., 2022) are variants of the soft $k$-NN loss introduced in Neighborhood Component Analysis (NCA) (Goldberger et al., 2004). We therefore explore a variant of our training setup where the loss directly minimizes the log NCA probabilities. Specifically, as in the OCM variant, we use a memory bank $\mathcal{Q}$ which stores $\ell_2$-normalized embeddings $\mathbf{z}$ output by the projector instead of using the full dataset to compute the soft $k$-NN, as the latter would be intractable. This variant then optimizes the following loss:

$$\mathcal{L}_{\text{OCA}} = -\frac{1}{M} \sum_{j=1}^{M} \sum_{c=1}^{C} \mathbf{y}_{[c]} \log \frac{\sum_{\mathbf{z}_c \in \mathcal{Q}_c} \exp(\mathbf{z}_j^\top \mathbf{z}_c / \tau)}{\sum_{\mathbf{z}_n \in \mathcal{Q}} \exp(\mathbf{z}_j^\top \mathbf{z}_n / \tau)}, \quad (3)$$

where $M$ is the number of crops, $C$ is the number of classes, $\mathbf{y} \in \{0, 1\}^C$ is the $C$-dim one-hot label vector, $\tau$ is temperature for NCA probabilities and $\mathcal{Q}_c$ denotes samples in the memory bank that belong to class $c$. Since this is an "online" variant of the NCA objective, we refer to this training objective as Online Component Analysis or OCA. Similar to OCM, we use momentum networks in OCA, i.e., momentum encoder $f_\xi$ and projector $g_\zeta$. For more details, see Appendix A.2.

We experimentally show in Fig. 10 that our t-ReX-OCA models achieve a balance between IN1K and transfer performances, i.e., they are towards the middle of the envelope in Fig. 10. Moreover, in our evaluations we see that this variant is better overall than LOOK+*multi-crop*. Note that the main difference between LOOK+*multi-crop* and the t-ReX-OCA variants is that the former restricts the soft k-NN loss (Goldberger et al., 2004) to the close neighborhood of each training sample in the memory bank. Our observation suggests that relying on more points in the memory bank is beneficial for learning better representations. We further observed that for this variant the size of the memory bank $|\mathcal{Q}|$ does not severely affect the performance of learned representations, but the smoothness of NCA probabilities computed in Eq. (3) does. Therefore, it is important to tune the temperature parameter $\tau$ for a given memory bank size to carefully control this smoothness.

## B  FURTHER DETAILS ON THE EVALUATION PROCESS

### B.1  HYPER-PARAMETERS FOR IN1K TRAINING

As we discuss in the previous section, we train supervised models on IN1K with different objectives and architecture configurations. We observe that the hyper-parameters for IN1K training that we share in Tab. 3 work well for the most effective models we studied.

### B.2  EVALUATION DATASETS

Once we train our models on IN1K, we evaluate their encoder representations $f_\theta(\mathbf{I})$ by training linear logistic regression (Log.Reg.) classifiers on 13 transfer datasets which include 5 ImageNet-CoG levels (Sariyildiz et al., 2021) and 8 commonly used small-scale datasets: Aircraft (Maji et al., 2013), Cars196 (Krause et al., 2013), DTD (Cimpoi et al., 2014), EuroSAT (Helber et al., 2019), Flowers (Nilsback & Zisserman, 2008), Pets (Parkhi et al., 2012), Food101 (Bossard et al., 2014) and SUN397 (Xiao et al., 2010). To test the generalization of models to IN1K concepts, we use the three test sets of IN1K-v2 (Recht et al., 2019) and out of domain images of IN1K-Sketch (Wang et al., 2019). Finally, to test performance on long-tail classification tasks, we use iNaturalist 2018 and 2019 (Van Horn et al., 2018). Statistics of all these datasets are provided in Tab. 4.

When a `val` split is not provided for a dataset, we randomly split its `train` set into two, following the size of `train` and `val` splits from either Feng et al. (2022) or Grill et al. (2020). We also created different `train`/`val` splits when tuning hyper-parameters with different seeds, thus further increasing the robustness of our scores. Other notes on the datasets are as follows:

  (i) For DTD (Cimpoi et al., 2014) (resp. EuroSAT (Helber et al., 2019)), there are 10 official `train`/`val`/`test` (resp. `train`/`test`) splits. Following Feng et al. (2022); Grill et al. (2020), we use the first split.
 (ii) For EuroSAT (Helber et al., 2019), we are not aware of either an official dataset split or the exact splits used in prior work, e.g., by Feng et al. (2022). So, we create random `train`/`val`/`test` splits following the number of samples in each split from Feng et al. (2022), ensuring that the `val` and `test` splits are balanced to contain the same number of samples for each class.
(iii) For iNaturalist 2018 and 2019 (Van Horn et al., 2018), we use the official `val` split as the `test` split and create a random `val` split each time with a different seed.
 (iv) We use the Log.Reg. classifier trained on IN1K for predicting image labels in the three test sets of IN1K-v2 or the test set of IN1K-Sketch. This is because IN1K-v2 and IN1K-Sketch are only composed of test sets and no training data is provided for them.

### B.3  EVALUATION METRICS: THE AVERAGE LOG ODDS TRANSFERABILITY SCORE

Following Kornblith et al. (2019), we compute log odds over all transfer datasets and use their average as a *transferability score*. This is the main metric we report in the different plots of the main paper. Denoting $n_{\text{correct}}$ and $n_{\text{incorrect}}$ as the number of correct and incorrect predictions for a dataset, we compute the accuracy $p$ and log odds score as follows:

$$p = \frac{n_{\text{correct}}}{n_{\text{correct}} + n_{\text{incorrect}}}, \quad \log \text{odds} = \log \frac{p}{1-p}. \tag{4}$$

Then we report log odds averaged over all transfer datasets. See Appendix C.2 for per-dataset top-1 accuracies and average log odd scores for the models we compare in the main paper.

Table 3: **Hyper-parameters for training** our models on IN1K. Hyper-parameters shared by all models are given on the top part while the ones specific to **t-ReX** and **t-ReX\*** are shown on the bottom part. Note that neither **t-ReX** nor **t-ReX\*** is trained with an expendable predictor head $h_\psi$.

| Configuration | Value for all models | |
|---|---|---|
| Optimizer | SGD | |
| Base learning rate | 0.1 | |
| Learning rate rule | $0.1 \times {}^{\text{batch size}}/_{256}$ | |
| Learning rate warmup | Linear, 10 epochs | |
| Learning rate decay rule | Cosine schedule | |
| Weight decay | 0.0001 | |
| Momentum | 0.9 | |
| Number of GPUs | 4 | |
| Batch size per GPU | 64 | |
| Batch size total | 256 | |
| Epochs | 100 | |
| Synchronized batch norms | ✓ | |
| Mixed precision | ✓ | |
| $\tau$ in Eqs. (1) and (2) | 0.1 | |
| Augmentation pipeline from | DINO | |
| Number of Global crops ($M_g$) | 1 | |
| Number of Local crops ($M_l$) | 8 | |
| Global crop resolution | 224 | |
| Global crop scale range | (0.4, 1) | |
| Local crop resolution | 96 | |
| Local crop scale range | (0.05, 0.4) | |
| | Value for **t-ReX** | Value for **t-ReX\*** |
| Projector input $\ell_2$-norm | ✓ | ✓ |
| Projector $L$ | 3 | 1 |
| Projector $d_h$ | 2048 | 2048 |
| Projector $d_b$ | 256 | 256 |
| Global crop scale range | (0.25, 1) | (0.4, 1) |
| Local crop scale range | (0.05, 0.25) | (0.05, 0.4) |
| Memory bank size $|\mathcal{Q}|$ | 8192 | 8192 |
| Loss function used for training | $\mathcal{L}_{\text{OCM}}$ | $\mathcal{L}_{\text{OCM}}$ |

## B.4 EVALUATION PROTOCOLS

We perform image classification on each evaluation dataset with logistic regression (Log.Reg.) classifiers following one of the two protocols proposed by Sariyildiz et al. (2021) (for the 5 CoG levels) or by Kornblith et al. (2019) (for the 8 small-scale datasets). In all cases, we first extract and store a (single) feature vector for each image and then learn the Log.Reg. classifiers on top of those features. Our classifiers are therefore trained *without data augmentation*, and this is why we report lower performance for the RSB model than the one presented by Wightman et al. (2021). We extract image representations from the encoders $f_\theta$ by resizing an image with bicubic interpolation such that its shortest side is 224 pixels and then taking a central crop of size $224 \times 224$ pixels. The protocols for training Log.Reg. classifiers on the 5 CoG levels and on the other 8 datasets are different and detailed below.

**Log.Reg. on the ImageNet-CoG levels.** We apply $\ell_2$-normalization to the pre-extracted features using the publicly-available source code of Sariyildiz et al. (2021), and then train Log.Reg. classifiers

Table 4: **Datasets** used for training (IN1K) and evaluating (the others) the quality of visual representations. We report top-1 accuracy for each dataset. Further implementation details on the utilization of the datasets are in Appendix A.1. *CCAS-4.0* denotes the Creative Commons Attribution-ShareAlike 4.0 international license.

| Dataset | # Classes | # Train samples | # Val samples | # Test samples | Val provided | Test provided | License |
|---|---|---|---|---|---|---|---|
| *For training models* | | | | | | | |
| IN1K | 1000 | 1281167 | – | 50000 | – | ✓ | Research-only |
| *For evaluating models on IN1K concepts* | | | | | | | |
| IN1K-v2 | 1000 | – | – | $3 \times 10000$ | – | ✓ | Research-only |
| IN1K-Sketch | 1000 | – | – | 50889 | – | ✓ | Research-only |
| *For evaluating models on transfer tasks* | | | | | | | |
| CoG $L_1$ | 1000 | 895359 | 223445 | 50000 | – | ✓ | Research-only |
| CoG $L_2$ | 1000 | 892974 | 222814 | 50000 | – | ✓ | Research-only |
| CoG $L_3$ | 1000 | 876495 | 218708 | 50000 | – | ✓ | Research-only |
| CoG $L_4$ | 1000 | 886013 | 221115 | 50000 | – | ✓ | Research-only |
| CoG $L_5$ | 1000 | 873630 | 218024 | 50000 | – | ✓ | Research-only |
| Aircraft | 100 | 3334 | 3333 | 3333 | ✓ | ✓ | Research-only |
| Cars196 | 196 | 5700 | 2444 | 8041 | – | ✓ | Research-only |
| DTD | 47 | 1880 | 1880 | 1880 | ✓ | ✓ | *Unclear* |
| EuroSAT | 10 | 13500 | 5400 | 8100 | – | – | Research-only |
| Flowers | 102 | 1020 | 1020 | 6149 | ✓ | ✓ | *Unclear* |
| Pets | 37 | 2570 | 1110 | 3669 | – | ✓ | CCAS-4.0 |
| Food101 | 101 | 68175 | 7575 | 25250 | – | ✓ | *Unclear* |
| SUN397 | 397 | 15880 | 3970 | 19850 | – | ✓ | Research-only |
| *For evaluating models on long-tail classification tasks* | | | | | | | |
| iNaturalist 2018 | 8142 | 437513 | – | 24426 | – | ✓ | Research-only |
| iNaturalist 2019 | 1010 | 265213 | – | 3030 | – | ✓ | Research-only |

using SGD with momentum = 0.9 and batch size = 1024 for 100 epochs. To treat each model as fairly as possible, we set the learning rate and weight decay hyper-parameters using `train`/`val` splits (`val` splits are randomly sampled using $20\%$ of the original `train` splits). We use Optuna (Akiba et al., 2019) and sample 30 different pairs. We train the final classifiers with these hyper-parameters on the union of the `train` and `val` splits, and report top-1 accuracy on the `test` splits. We repeat this process 5 times with different seeds and report averaged results.

**Log.Reg. on the smaller-scale transfer datasets.** Following Kornblith et al. (2019), we train Log.Reg. classifiers with pre-extracted features using L-BFGS (Liu & Nocedal, 1989). To this end, we use the implementation in Scikit-learn (Pedregosa et al., 2011). We set the inverse regularization coefficient ("C") on each dataset using Optuna and their `train`/`val` splits over 25 trials. If a dataset does not have a fixed validation set, then we repeat hyper-parameter selection 5 times with different seeds and report the average result.

### B.5 HYPER-PARAMETERS FOR **T-REX** AND **T-REX*** ON TRANSFER DATASETS

In Tab. 5 we present the hyper-parameters found by Optuna (Akiba et al., 2019) for the Log.Reg. classifiers trained on the transfer datasets for **t-ReX** and **t-ReX***.

### B.6 LIST OF PUBLICLY AVAILABLE PRETRAINED MODELS USED FOR COMPARISONS

In Sec. 4.3 of the main paper, we compare our models to several prior works which are state-of-the-art either for IN1K classification or for transfer learning. These include self-supervised DINO (Caron et al., 2021), semi-supervised PAWS (Assran et al., 2021) and supervised SupCon (Khosla et al., 2020), RSB-A1 (Wightman et al., 2021), LOOK (Feng et al., 2022) and SL-MLP (Wang et al., 2022b). Additionally, we evaluated another self-supervised model Barlow Twins (Zbontar et al., 2021) but found it to be inferior to DINO on both IN1K and transfer datasets. For all the models except LOOK and SL-MLP, we evaluate the best ResNet50 encoders trained on IN1K by their respective authors. Since there was neither publicly-available model nor source code for these two models, we reproduced the methods ourselves and found that they can perform significantly better when

Table 5: **Hyper-parameters for evaluating** our **t-ReX** and **t-ReX\*** models using Log.Reg. classifiers on transfer datasets found by Optuna (Akiba et al., 2019). "Learning rate" and "Weight decay" are the parameters of the SGD optimizer used for training Log.Reg. classifiers as in the ImageNet-CoG protocol Sariyildiz et al. (2021). $C$ is the inverse regularization coefficient used when training Log.Reg. classifiers with L-BFGS implemented in Scikit-learn (Pedregosa et al., 2011).

| | Configuration | $IN1K$ | $CoG\ L_1$ | $CoG\ L_2$ | $CoG\ L_3$ | $CoG\ L_4$ | $CoG\ L_5$ | $Aircraft$ | $Cars196$ | $DTD$ | $EuroSAT$ | $Flowers$ | $Pets$ | $Food101$ | $SUN397$ |
|---|---|---|---|---|---|---|---|---|---|---|---|---|---|---|---|
| t-ReX | Learning rate | 12.8 | 11.0 | 10.8 | 11.0 | 11.0 | 11.0 | – | – | – | – | – | – | – | – |
| | Weight decay | 2.4e-10 | 1.1e-8 | 7.3e-10 | 1.1e-8 | 1.1e-8 | 1.1e-8 | – | – | – | – | – | – | – | – |
| | C | – | – | – | – | – | – | 42169 | 5109 | 1.1 | 9.1 | 14678 | 1778 | 0.4 | 1.1 |
| t-ReX\* | Learning rate | 16.9 | 23.6 | 28.6 | 21.4 | 75.3 | 75.3 | – | – | – | – | – | – | – | – |
| | Weight decay | 1.2e-8 | 6.3e-10 | 4.3e-9 | 1.8e-9 | 4e-8 | 4e-8 | – | – | – | – | – | – | – | – |
| | C | – | – | – | – | – | – | 42169 | 14667 | 1.1 | 75 | 14678 | 42169 | 3.2 | 3.2 |

Table 6: **Compared models** with public ResNet50 encoder weights trained on IN1K by the authors.

| Model | Epochs | Additional Notes | Repository URL and License |
|---|---|---|---|
| DINO (Caron et al., 2021) | 800 | Self-supervised | https://github.com/facebookresearch/dino Apache-2.0 |
| PAWS (Assran et al., 2021) | 300 | Semi-supervised (With $10\%$ of annotations) | https://github.com/facebookresearch/suncet MIT |
| SupCon (Khosla et al., 2020) | 800 | Supervised (with momentum encoder and memory bank) | https://github.com/HobbitLong/SupContrast BSD-2-Clause |
| RSB-A1 (Wightman et al., 2021) | 600 | Supervised | https://github.com/rwightman/pytorch-image-models Apache-2.0 |

combined with multi-crop. We compare to these enhanced version which we call LOOK+*multi-crop* and SL-MLP+*multi-crop*. In Tab. 6, we give a list of the compared models with public encoder weights.

## C   Extended results and evaluations

### C.1   Results per dataset

In Tab. 7, we report top-1 accuracy on each transfer dataset and on IN1K. Results are obtained by Log.Reg. classifiers, for all the models listed in Tab. 6 and for the ones that belong to the "convex hull" or envelope, denoted by stars in Fig. 7 of the main paper.

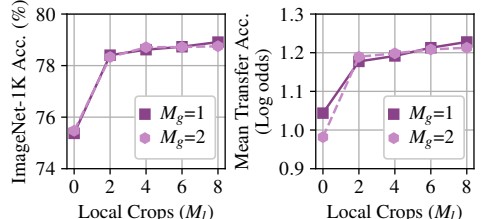
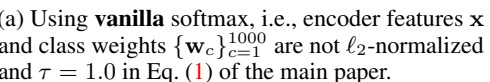
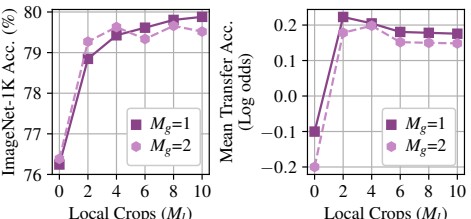

(a) Using **vanilla** softmax, i.e., encoder features $\mathbf{x}$ and class weights $\{\mathbf{w}_c\}_{c=1}^{1000}$ are not $\ell_2$-normalized and $\tau = 1.0$ in Eq. (1) of the main paper.

(b) Using **cosine** softmax, following Eq. (1) of the main paper, i.e., encoder features $\mathbf{x}$ and class weights $\{\mathbf{w}_c\}_{c=1}^{1000}$ are $\ell_2$-normalized and $\tau = 0.1$.

Figure 11: **Ablating the number of global and local crops for different softmax losses** without using a projector head, i.e., $g_\phi$ is an identity mapping in Eq. (1) of the main paper.

Table 7: **Top-1 Log.Reg. accuracy per dataset.** Mean LO is average log odds computed over all transfer datasets (i.e., all datasets except IN1K). In the main paper, we only plot IN1K and Mean LO scores for each model. For the datasets without a fixed validation set (see Appendix B.2) we repeat each evaluation 5 times with different seeds; variance is generally negligible.

| Model | IN1K | CoG $L_1$ | CoG $L_2$ | CoG $L_3$ | CoG $L_4$ | CoG $L_5$ | Aircraft | Cars196 | DTD | EuroSAT | Flowers | Pets | Food101 | SUN397 | Mean LO |
|---|---|---|---|---|---|---|---|---|---|---|---|---|---|---|---|
| *Previous SotA* | | | | | | | | | | | | | | | |
| DINO (Caron et al., 2021) | 74.8 | 71.1 | 67.2 | 63.2 | 62.6 | **57.6** | 62.5 | 67.4 | **77.7** | **97.7** | 95.6 | 88.9 | 78.7 | 66.0 | 1.256 |
| PAWS (Assran et al., 2021) | 76.4 | 71.2 | 67.3 | 63.1 | 62.1 | 56.6 | 63.2 | 71.6 | 76.2 | 96.9 | 95.8 | 91.2 | 77.5 | 65.4 | 1.256 |
| SL-MLP (Wang et al., 2022a) | 75.1 | 70.1 | 66.1 | 61.6 | 60.4 | 54.5 | 63.1 | 70.9 | 75.0 | 96.7 | 94.8 | 91.6 | 74.9 | 63.7 | 1.189 |
| LOOK+*multi-crop* (Feng et al., 2022) | 78.0 | 70.2 | 65.9 | 61.7 | 60.4 | 54.7 | 62.4 | 71.1 | 73.5 | 96.3 | 94.9 | 93.3 | 75.1 | 64.1 | 1.195 |
| SupCon (Khosla et al., 2020) | 78.8 | 69.9 | 64.7 | 60.6 | 59.1 | 53.1 | 57.3 | 60.9 | 74.6 | 95.7 | 91.6 | 92.8 | 71.9 | 62.8 | 1.053 |
| RSB-A1 (Wightman et al., 2021) | 79.8 | 69.9 | 65.0 | 60.9 | 59.3 | 52.8 | 47.1 | 54.0 | 73.9 | 95.7 | 88.7 | 93.1 | 71.2 | 63.3 | 0.978 |
| *Our models on the convex hull in Fig. 7 of the main paper* | | | | | | | | | | | | | | | |
| **t-ReX** | 78.0 | 72.0 | **68.3** | **63.9** | **63.4** | 57.2 | **67.3** | **74.2** | **77.7** | 97.5 | **96.2** | 92.6 | **80.1** | 66.7 | **1.357** |
| t-ReX-OCM ($L$=2, $\mathcal{Q}$=8K) | 78.8 | **72.3** | 68.2 | 63.7 | 63.0 | 56.8 | 64.7 | 70.8 | 75.8 | 97.3 | 95.3 | 93.2 | 79.1 | **66.9** | 1.305 |
| t-ReX-OCM ($L$=1, $h_\psi$, $|\mathcal{Q}|$=131K) | 79.6 | 71.7 | 67.3 | 62.8 | 61.6 | 55.3 | 61.9 | 68.8 | 75.2 | 96.7 | 94.0 | **93.6** | 76.6 | 66.1 | 1.224 |
| t-ReX$_1$ ($\ell_2$, $L$=1, $d_h$=4096, $d_b$=256) | 79.8 | 71.7 | 67.1 | 63.0 | 61.8 | 54.8 | 61.1 | 66.7 | 74.4 | 96.8 | 93.2 | 93.5 | 76.7 | 66.2 | 1.201 |
| t-ReX$_1$ ($\ell_2$, $L$=1, $d_h$=2048, $d_b$=256) | 80.0 | 71.3 | 66.4 | 62.3 | 60.6 | 53.9 | 58.8 | 67.5 | 75.2 | 96.4 | 91.6 | 93.4 | 75.4 | 65.4 | 1.150 |
| **t-ReX*** | **80.2** | 70.7 | 66.0 | 61.5 | 59.8 | 53.4 | 55.5 | 64.7 | 73.2 | 96.2 | 90.1 | 93.0 | 73.2 | 64.8 | 1.078 |

Table 8: **Varying the scale and resolution of global and local crops as well as the batch size.** We train models using different minimum and maximum scales for global and local crops, and batch size. $M_l$ is the number of local crops. We use 1 global crop, i.e., $M_g = 1$, for each experiment. PyTorch, Simsiam and DINO augmentation pipelines are from Paszke et al. (2019), Chen & He (2021) and Caron et al. (2021), respectively. Note that for the experiments presented in this table we train models using vanilla softmax, see Appendix C.3 for a discussion.

| | Augmentation Pipeline | Global Scale | Local Scale | Local Resolution | $M_l$ | Epoch | Batch Size | IN1K Top-1 | Mean Transfer Log odds |
|---|---|---|---|---|---|---|---|---|---|
| 1 | PyTorch | (0.08, 1.00) | – | – | – | 100 | 256 | 76.0 | 1.07 |
| 2 | SimSiam | (0.20, 1.00) | – | – | – | 100 | 256 | 76.0 | 1.06 |
| 3 | DINO | (0.05, 1.00) | – | – | – | 100 | 256 | 76.4 | 1.08 |
| 4 | DINO | (0.05, 1.00) | – | – | – | 100 | 2304 | 76.5 | 1.07 |
| 5 | DINO | (0.05, 1.00) | – | – | – | 800 | 256 | 76.5 | 1.13 |
| 6 | DINO | (0.15, 1.00) | (0.05, 0.15) | $96 \times 96$ | 8 | 100 | 256 | 78.4 | 1.19 |
| 7 | DINO | (0.25, 1.00) | (0.05, 0.25) | $96 \times 96$ | 8 | 100 | 256 | 78.6 | 1.21 |
| 8 | DINO | (0.40, 1.00) | (0.05, 0.40) | $96 \times 96$ | 8 | 100 | 256 | **78.9** | **1.23** |
| 9 | DINO | (0.40, 1.00) | (0.05, 0.40) | $224 \times 224$ | 2 | 100 | 256 | 77.5 | 1.08 |
| 10 | DINO | (0.40, 1.00) | (0.05, 0.40) | $192 \times 192$ | 2 | 100 | 256 | 77.5 | 1.10 |
| 11 | DINO | (0.40, 1.00) | (0.05, 0.40) | $160 \times 160$ | 2 | 100 | 256 | 77.8 | 1.13 |
| 12 | DINO | (0.40, 1.00) | (0.05, 0.40) | $128 \times 128$ | 2 | 100 | 256 | 78.1 | 1.17 |
| 13 | DINO | (0.40, 1.00) | (0.05, 0.40) | $96 \times 96$ | 2 | 100 | 256 | 78.3 | 1.19 |
| 14 | DINO | (0.40, 1.00) | (0.05, 0.40) | $64 \times 64$ | 2 | 100 | 256 | 77.8 | 1.17 |
| 15 | DINO | (0.40, 1.00) | (0.05, 0.40) | $32 \times 32$ | 2 | 100 | 256 | 74.8 | 1.02 |

## C.2 EXTENDED VERSION OF FIG. 7

In Fig. 10 we present an extended version of Fig. 7 in the main paper that further includes results for the variants presented in Appendices A.3 and A.4.

## C.3 EXTENDED ABLATIONS ON MULTI-CROP

In Sec. 4.1 of the main paper, we ablate the hyper-parameters of multi-crop using a projector head. This is because cosine softmax loss, when used without a projector head, suffers from overfitting to IN1K, and yields much worse transfer performance, as we show in Fig. 11. Note that this phenomenon has also been observed by Kornblith et al. (2021) and explained by the fact that cosine softmax increases class separability of seen concepts in the feature space which reduces transferability. Therefore, we believe that using cosine softmax loss when training "multi-crop-only" models is clearly sub-optimal to set multi-crop hyper-parameters. However, we note that, as shown in Sec. 4 of the main paper, this overfitting of cosine softmax is alleviated by projector heads. In the remaining of

Table 9: **Extended results for the effect of the projector's design.** We start from a default configuration, i.e., $L = 1$, $d_h = 2048$, $d_b = 256$ and with $\ell_2$ normalization of the input (corresponding to highlighted rows in the tables), and ablate $L$, $d_b$ and $\ell_2$-normalization separately, training models with Eq. (1). These two tables complement Tab. 1 of the main paper.

(a) Bottleneck dimension $d_b$

| $d_b$ | IN1K | Transfer |
|-----|------|----------|
| 128 | 79.8 | 1.14 |
| 256 | 79.8 | 1.15 |
| 512 | 79.9 | 1.16 |
| 1024 | 79.6 | 1.15 |
| 2048 | 80.0 | 1.15 |

(b) Hidden layer $L$ and input $\ell_2$-norm

| $L$ | $\ell_2$ | IN1K | Transfer |
|-----|------|------|----------|
| 1 | – | 79.4 | 1.17 |
| 1 | ✓ | 79.8 | 1.15 |
| 2 | – | 78.6 | 1.33 |
| 2 | ✓ | 78.6 | 1.31 |
| 3 | – | 77.9 | 1.35 |
| 3 | ✓ | 77.5 | 1.33 |

this section, we train "multi-crop-only" models using vanilla softmax, i.e., encoder features $\mathbf{x}$ and class weights $\{\mathbf{w}_c\}_{c=1}^{1000}$ are not $\ell_2$-normalized and $\tau = 1.0$ in Eq. (1) of the main paper.

In the main paper, we use (0.05, 0.4) and (0.4, 1.0) as the scale range for local and global crops, i.e., we sample scale values from these intervals. Tab. 8 provides an ablation on the maximum and minimum scales for local and global crops, respectively (see rows 6-8 in Tab. 8). We evaluate 3 different values (0.15, 0.25 and 0.4) for the maximum scale of local crops, which is also set as the minimum scale for global crops. Although the results are comparable, we see that 0.40 produces slightly better results.

We further verify if the improvements from local crops are due to the fact that a) models are trained using *more* variants for each image, or b) the effective batch size is increased during training (as we use one batch of images for each crop). To test a), we train a model for 800 epochs using single crops with scale range (0.05, 1.0), and observe that (see row 5 in Tab. 8) it performs comparably to training models without multi-crop, i.e., it barely improves over training models with PyTorch and SimSiam augmentations (rows 1 and 2) and performs similar when DINO augmentations are used without multi-crop (row 3). To test b), we train a model for 100 epochs using single crops with scale range (0.05, 1.0) but increase batch size by $9\times$ (see row 4). Yet, we again observe no improvements over training models without local crops.

Finally, we ablate the resolution of local crops (rows 9-15 in Tab. 8). Using two local crops $M_l = 2$ and a single global crop $M_g = 0$, we try several values as for the resolution of local crops $[224 \times 224, 192 \times 192, 160 \times 160, 128 \times 128, 96 \times 96, 64 \times 64, 32 \times 32]$ and find that $96 \times 96$ works best on both IN1K and transfer datasets.

## C.4 EXTENDED ABLATIONS ON THE PROJECTOR DESIGN

We present additional ablations on the projector design, complementing our discussion in Sec. 4.1 of the main paper. We analyse the effect of the bottleneck layer dimension ($d_b$), and whether or not to $\ell_2$-normalize the projector input ($\ell_2$) to the performance on IN1K and transfer tasks. As we see in Tab. 9, they have only a small influence on the performance.

## C.5 EXTENDED ABLATIONS FOR ONLINE CLASS MEANS (OCM)

**Varying the projector and predictor head architecture.** In Tab. 1 of the main paper, we study the impact of the projector head parameters for the models we train with Eq. (1). Here, we replicate this study for the OCM variant. Specifically, we ablate the $L$ and $d_h$ parameters for the OCM model defined by Eq. (2). We show the results in Fig. 12a, and see that our observation still holds, i.e., increasing the complexity of projector improves transfer performance at the cost of IN1K performance. We also tested the impact of predictors in OCM models, and observed a similar behavior: Small predictors improve the transferability of encoder representations by sacrificing IN1K performance; allowing these variants to move along the envelope. Consequently, training OCM models with no predictor, i.e., when $h_\psi$ is an identity mapping in Fig. 9b, improves IN1K performance. Indeed, our best model on IN1K, i.e., **t-ReX\***, is a **t-ReX$_1$-OCM** variant with no predictor head.

**Varying the memory bank size.** Fig. 12b shows how the size of the memory bank impacts performance for the OCM variant, where we see that even a moderately-sized memory bank of $|\mathcal{Q}| = 8192$, i.e., containing only 8 points per class on average is sufficient to obtain high performance on both axes.

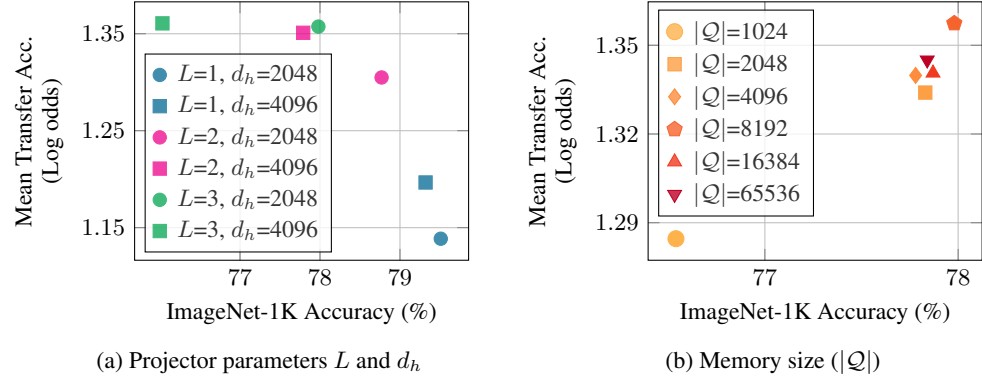

(a) Projector parameters $L$ and $d_h$  (b) Memory size ($|\mathcal{Q}|$)

Figure 12: **OCM ablations** for (a) the projector head parameters $L$ and $d_h$ when $|\mathcal{Q}| = 8192$ and (b) the size of memory bank $|\mathcal{Q}|$ when $L = 3$ and $d_h = 2048$.

## C.6 EXTENDED ANALYSES OF LEARNED FEATURES, CLASS WEIGHTS AND PROTOTYPES

In Figs. 5 and 6 of the main paper, we compare six models: a) *Base*: a model trained using cosine softmax loss without multi-crop and projector, b) *Base (BS=2K)*: Base but with $9\times$ larger batch size, c) *Base+Mc*: Base plus multi-crop, d) *Base+Pr*: Base plus a projector, e) *Base+Mc+Pr*, and f) *OCM*: a model trained using Eq. (2), from multiple aspects to get more insights on how our proposed training setup improves performance. We extend these analyses below.

When comparing models, we use either features extracted using these models on each dataset, their class weights, or their prototypes. For the analyses with features, we use only the samples in the `test` splits of IN1K and CoG levels, but all the samples in the others (as these datasets are small-scale).

**Pairwise $\ell_2$-distance.** We $\ell_2$-normalize features and compute average pairwise $\ell_2$-distance either between samples from the same class (intra-class) or between all samples in a dataset (all-sample). In Fig. 5 (left) of the main paper, we show intra-class distance for the four models (a, c, d and e) and for completeness, in Fig. 13 (left), we provide all-sample $\ell_2$-distances for them.

**Sparsity.** We compute sparsity as the percentage of feature dimensions close to zero. To compute this metric, we apply a threshold $\epsilon$ to individual dimensions of $\ell_2$-normalized features. In Fig. 5 (right) of the main paper, we show the results obtained with $\epsilon = 10^{-5}$, but we found that the conclusions obtained with a range of $\epsilon$ values logarithmically sampled between $10^{-3}$ and $10^{-8}$ were consistent.

**Coding length.** We compute average coding length (Ma et al., 2007; Yu et al., 2020) of samples in a dataset as $R(\mathbf{X}, \epsilon) = \frac{1}{2}\log\det(\boldsymbol{I}_d + \frac{d}{N\epsilon^2}\mathbf{X}^\top\mathbf{X})$, where $\boldsymbol{I}_d$ is a $d$-by-$d$ identity matrix, $\epsilon^2$ is the precision parameter set to 0.5 and $\mathbf{X} \in \mathbb{R}^{N \times d}$ is a feature matrix containing $N$ samples each encoded into a $d$-dimensional representation (2048 in our case). In Fig. 6 (left) of the main paper, we show average coding length for the four models (a, c, d and e) obtained on transfer datasets (i.e., coding length values are further averaged over transfer datasets). In Fig. 14 we show coding length for the same four models plus DINO Caron et al. (2021) obtained on each dataset, separately. Moreover, we compute singular values per dimension for models with and without projectors (Base+Pr and Base+Mc). For each model, we compute singular values on each transfer dataset which are normalized by their sum so that they sum to 1. We then sort these normalized singular values by decreasing order, and average them over transfer datasets. As can be seen in Fig. 13 (right), feature variance is more uniformly distributed over dimensions when a projector is used.

**Gradient similarity and variance.** To get a better understanding on why multi-crop improves performance significantly for the same effective batch size, we compare the models Base+Mc and Base (BS=2K) by analyzing gradients of their class weights $\nabla_{\mathbf{W}}\mathcal{L}_{\text{CE}}$ at each SGD update. We compute absolute cosine similarity between "class gradients" for a pair of classes $c_i$ and $c_j$ as $|\text{sim}(\nabla_{\mathbf{w}_{c_i}}\mathcal{L}_{\text{CE}}, \nabla_{\mathbf{w}_{c_j}}\mathcal{L}_{\text{CE}})|$, where $\text{sim}(\cdot, \cdot)$ denotes cosine similarity. Then we average them over all class pairs $(c_i, c_j) \sim \mathcal{P}$ in a dataset, which is reported in Fig. 6 (middle) of the main paper. Additionally, in Fig. 16 we show Frobenius norm and standard deviation computed over $\nabla_{\mathbf{W}}\mathcal{L}_{\text{CE}}$. We observe that norm and variance of gradients increase with multi-crop, which we believe is due to increasing variance in embeddings as noted by Wang et al. (2022a).

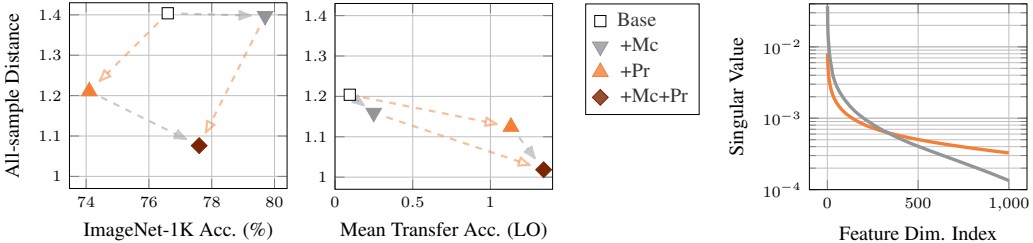

Figure 13: (*left*) **All-sample $\ell_2$ distance** vs. performance on IN1K and transfer datasets. These two sub-plots extend the intra-class distance plots shown in Fig. 5 (left) of the main paper. Gray and Orange arrows denote changes due to adding multi-crop and projectors, respectively. (*right*) **Singular values** across dimensions, averaged over the transfer datasets. We show the first 1000 dimensions (of 2048) for clarity.

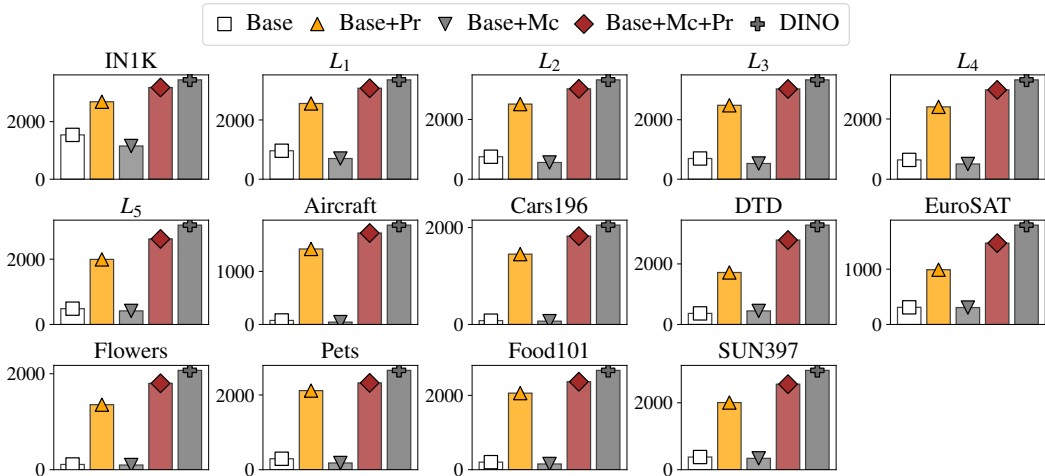

Figure 14: **Average coding length** (Yu et al., 2020) obtained on each dataset separately. See Appendix C.6 for a description of models. In Fig. 5 (middle) of the main paper, we show average coding length over the transfer datasets (excluding IN1K) for all the models except DINO.

**Feature redundancy.** In addition to the metrics described above, following Wang et al. (2022b), we compute "redundancy" of features $\mathbf{X} \in \mathbb{R}^{N \times d}$ (defined as in the "coding length" paragraph above) as $\mathcal{R} = \frac{1}{d^2} \sum_i \sum_j |\rho(\mathbf{X}_{:,i}, \mathbf{X}_{:,j})|$, where , $\rho(\mathbf{X}_{:,i}, \mathbf{X}_{:,j})$ is the Pearson correlation between a pair of feature dimensions $i$ and $j$. In Fig. 15 we show redundancy score for the four models (a, c, d and e) plus DINO Caron et al. (2021) obtained on each dataset, separately.

## C.7 RESULTS ON IN1K-V2 AND IN1K-SKETCH

We compare **t-ReX** and **t-ReX\*** to the previous state of the art on IN1K-v2 and IN1K-Sketch. As before, for each model, we use the trained encoder as a feature extractor, and we reuse the linear classifier trained on IN1K and apply it directly to the test images of IN1K-v2 and IN1K-Sketch. Note that there are 3 test sets for IN1K-v2, and we evaluate over all of them. Tab. 10 presents our results. Looking at the mean top-1 accuracy over the three test sets of IN1K-v2, we observe that **t-ReX\*** also matches the performance of RSB-A1 on IN1K-v2 outperforming all others, showing strong domain generalization capabilities. On the other hand, SupCon performs the best on IN1K-Sketch, where **t-ReX\*** is the second best. We think that the contrastive loss used in SupCon might have improved its out-of-distribution robustness for the training concepts.

## C.8 RESULTS ON CLASS-IMBALANCED TRANSFER DATASETS

We evaluate the long-tail transfer classification performance of DINO, PAWS, RSB-A1, **t-ReX** and **t-ReX\*** on two class-imbalanced datasets, iNaturalist 2018 and iNaturalist 2019 (Van Horn et al., 2018). For these evaluations, we follow the Log.Reg. protocol from the ImageNet-CoG benchmark

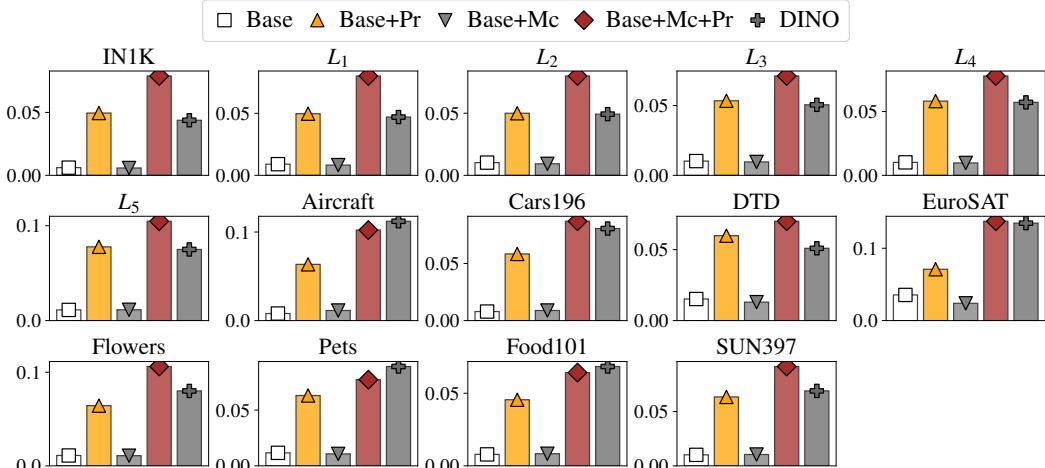

Figure 15: **Feature redundancy scores** (Wang et al., 2022b) obtained on each dataset separately. See Appendix C.6 for a description of models.

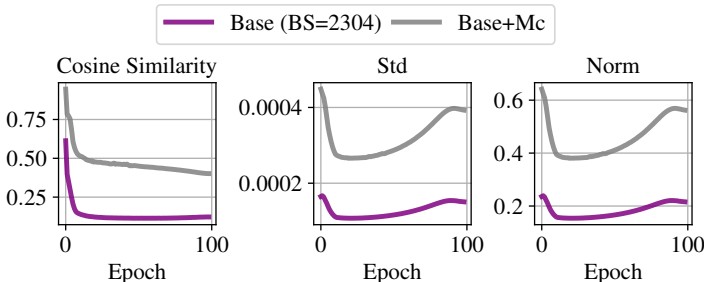

Figure 16: (*left*) Average cosine similarity between gradients of class weights $\nabla_{\mathbf{w}_c}\mathcal{L}_{\text{CE}}$ (also shown in Fig. 6 (middle) of the main paper). (*middle and right*) Standard deviation and Frobenius norm, respectively, computed over gradients $\nabla_{\mathbf{W}}\mathcal{L}_{\text{CE}}$. See Appendix C.6 for a description of models.

Table 10: **Results on IN1K concepts.** For each model, we report results on the original IN1K "Val" set (the x-axis of Fig. 7), as well as on the test sets of IN1K-sketch (Wang et al., 2019) and IN1K-v2 (Recht et al., 2019), using in all cases the encoder, and the linear classifier trained on the original IN1K training set.

| Model | IN1K Val | IN1K Sketch | IN1K-v2 Matched-frequency | Threshold-0.7 | Top-images | Mean |
|---|---|---|---|---|---|---|
| DINO | 74.8 | 19.8 | 61.9 | 71.2 | 76.7 | 69.9 |
| PAWS | 76.4 | 24.2 | 63.6 | 73.0 | 78.3 | 71.6 |
| LOOK + *multi-crop* | 78.0 | 27.8 | 65.8 | 75.3 | 80.7 | 73.9 |
| SupCon | 78.8 | **30.9** | 66.8 | 75.5 | 80.5 | 74.3 |
| RSB-A1 | 79.8 | 27.9 | 68.1 | 76.6 | 81.6 | 75.4 |
| **t-ReX** | 78.0 | 26.8 | 65.6 | 74.9 | 80.2 | 73.6 |
| **t-ReX\*** | **80.2** | 29.1 | **69.0** | **77.5** | **82.0** | **76.2** |

Table 11: **Transfer results on long-tail classification.** For each model, we train linear classifiers on the iNaturalist 2018 and iNaturalist 2019 datasets (Van Horn et al., 2018) with class-imbalanced data.

| Model | iNaturalist 2018 | iNaturalist 2019 |
|-------|------------------|------------------|
| DINO | 41.9 | 51.4 |
| PAWS | 40.8 | 49.8 |
| RSB-A1 | 34.9 | 43.2 |
| **t-ReX** | **45.8** | **54.2** |
| **t-ReX\*** | 36.0 | 44.2 |

(see Appendix B.4). Results are reported in Tab. 11. We see that our **t-ReX** and **t-ReX\*** models still outperform RSB-A1 and DINO respectively, despite a challenging long-tail class distribution.

## D  LIMITATIONS

**Requirement for annotations.** Firstly, our training setup is tailored for supervised learning, and therefore, its performance on both training and transfer tasks depends on the availability of high-quality and diverse annotations for the training images. Image-level annotation, when involving a large number of potentially fine-grained classes is an expensive and error-prone process. In this work, we show that given a large-scale *curated* and annotated dataset, more precisely given IN1K, which is composed of 1.28M images annotated for 1000 different concepts, it is possible to learn more generic representations than self- or semi-supervised models. When only a handful of concepts is annotated, or when annotations contain too much noise, these conclusions might not be accurate anymore, and both the pretraining classification task and the transfer tasks results might be degraded. In this case, self- or semi-supervised approaches might become more relevant. Yet, those scenarios are out of the scope of our study.

**Specific encoder architecture.** Secondly, we develop our training setup based on a single encoder architecture, ResNet50, and do not test it on other architectures. This was motivated by the fact that ResNet50 is still a very commonly used architecture. Also, we note that our training setup components, multi-crop, a data-augmentation operator, and the expendable projector, added after the encoder, are architecture-agnostic so those contributions can be seamlessly applied to any other architecture of choice. Therefore, it is reasonable to expect that our setup would consistently improve other architecture families, such as Vision Transformers (ViTs) Dosovitskiy et al. (2021). In fact, both components were previously successfully used with ViTs for self-supervised learning Caron et al. (2021). We leave studying the applicability of our training setup to other encoder architectures as future work.

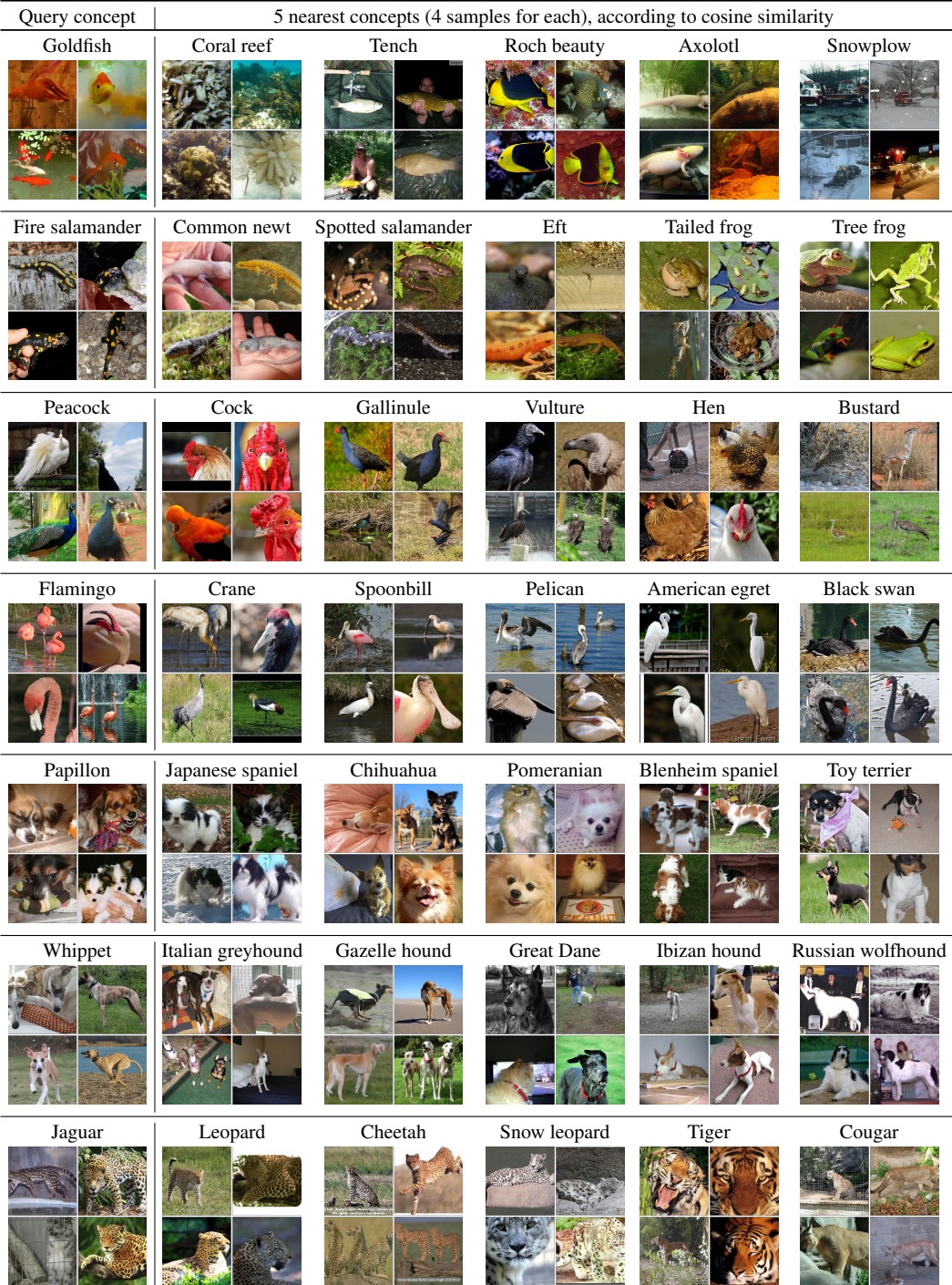

Table 12: **Visualization of the nearest neighbors of random class prototypes for t-ReX\*.** For each row, the first column is the query concept and the next 5 columns are the closest 5 other concepts according to cosine distance of their prototypes. All concepts are from IN1K and we show their images in the validation set.

