# OpenReview forum: "No Reason for No Supervision: Improved Generalization in Supervised Models"
_ICLR.cc/2023/Conference — ICLR 2023 notable top 25%_

### Official Review · Reviewer_p7Yg · 2022-10-15

**Confidence:** 4
**Correctness:** 4
**Technical Novelty And Significance:** 2
**Empirical Novelty And Significance:** 3
**Recommendation:** 8

**Clarity, Quality, Novelty And Reproducibility:**

The writing is very good and clear, as is the organization of the paper. The summarization the findings makes things very clear for the reader.


**Strength And Weaknesses:**

### Strengths (S) and weaknesses (W)

- S1: Interesting setting: focus on the *trade-off* between performance on training task / transfer tasks.
- S2: The empirical evaluation is extensive and do adequately support the claims.
- S3: The proposed enhancements to a supervised model lead to two SOTA models on each of the two settings, with competitive performance on the other respectively.

- (W1) This is mostly an empirical/engineering paper. A lot of different pieces are put together and evaluated empirically. There is surely value in these results and SOTA models, but I am not sure what findings are applicable/relevant beyond IN-1k. Can the authors comment on this?
- (W2) What does "cosine" mean in "cosine softmax"? The only difference with a standard softmax seems to be a summation over crops, and an additional vector of learned class-wise weights, that follow an L2 normalization of the outputs.

**Summary Of The Paper:**

This paper is about the performance of models trained on ImageNet-1K. The authors take the premise that performance on IN-1K and on other transfer tasks are desirable, and they examine how to achieve these two objectives simultaneously. They have traditionally been achieve distinctly by supervised and self-supervised approaches, respectively. The main concrete contribution is a suite of enhancements to a supervised model.

**Summary Of The Review:**

Not sure about the scientific value of this paper. But the contributions should nevertheless be of interest to some of the audience, and I can see no obvious flaws. Therefore I recommend acceptance!

---

> ### Author Response · Authors · 2022-11-16
> **Author's Response**
>
> > **What findings are applicable or relevant beyond ImageNet-1K?**
>
> This is indeed an open question.
>
> We acknowledge that our entire analysis is built on ImageNet-1K. As it is the most standard dataset for visual representation learning, we can directly and fairly compare all recent contributions in this field.
>
> Yet, to partially address this comment, we trained our t-ReX model on the full ImageNet dataset (using the images for the 19K concepts available in [the last release of ImageNet](https://image-net.org/update-mar-11-2021.php)).
>
> The table below reports preliminary results, following the evaluation protocol introduced in the paper, for two models: t-ReX as already defined in the paper (trained on ImageNet-1K), and a new t-ReX flavor trained on the full ImageNet dataset (ImageNet-19K). Note that, since the ImageNet-CoG datasets are now part of the training set, we exclude them when computing transfer performance.
>
> | Model | Training Dataset | ImageNet-1K (Top-1 Accuracy) | Average Transfer (Log odds, excluding CoG datasets) |
> |---|---|:---:|:---:|
> | t-ReX | ImageNet-1K | 78.0 | 1.8056 |
> | t-ReX | ImageNet-19K | 78.5 | 2.1506 |
>
> We observe stronger transfer learning performance when training on a superset of ImageNet-1K, i.e. a much larger dataset.
> We would like to stress that, as already discussed in a response to R-63Ba, running comparisons to DINO, PAWS, SupCon, LOOK, RSB on any additional dataset would require a prohibitive amount of compute.
>
> > **What does “cosine” mean in “cosine softmax”?**
>
> For “cosine softmax”, instead of using the dot product between embeddings and class weights (or prototypes) inside the softmax function of the cross entropy loss, we use the cosine similarity. In practice, this means that both the embeddings and the class weights (or prototypes) are L2-normalized vectors.

---

> > ### Comment · Reviewer_p7Yg · 2022-11-16
> > **Happy with response**
> >
> > I'm happy with the authors' response. I think that the novelty is limited, but the paper should nevertheless be of interest to some of the ICLR's audience.

---

### Official Review · Reviewer_yEpr · 2022-10-23

**Confidence:** 4
**Correctness:** 3
**Technical Novelty And Significance:** 1
**Empirical Novelty And Significance:** 2
**Recommendation:** 5

**Clarity, Quality, Novelty And Reproducibility:**

The writing is clear but with much redundant information. For example, the introduction section uses 1 page of the space but essentially the message is that the authors try to combine 3 well-known components in a supervised learning setting. I encourage the authors to spend more space on the analysis of class-imbalanced downstream tasks and show whether supervised learning still can outperform self-supervised learning for transfer learning.

**Strength And Weaknesses:**

Strength:
- The proposed modifications are straightforward and easy to follow. They are easily reproducible as most of the components have been released by previous works.
- Various figures (2 & 3) are helpful in comprehending the proposed changes.
- I appreciate the ablation study of those 3 components with detailed performance analysis.

Weaknesses
- As discussed in related work by the authors themselves, all 3 components have been proven useful. The multi-crop data augmentation and expendable projector head are now widely used in various self-supervised learning approaches. Directly applying and combining 3 well known components in my opinion does not necessarily guarantee meeting the novelty bar at ICLR.
- For example, since AlexNet, VGG, and ResNet, various different image cropping methods have been proposed. With the recent semi-supervised learning advances, we now know that more aggressive image cropping can generally make the training task harder, making the network learn harder and generalize better. The authors found that multi-crop data augmentation from [Caron et al 2020] also works well on ResNet under supervised learning setting should not be a huge surprise.
- Also, expendable projector head essentially forces the downstream tasks to use feature representations from the earlier layer (not the last layer) has also been largely explored such as [A1, A2]. One can essentially use any combination of any previous layers for the downstream task training.
- The major challenge in transferring features learned from supervised learning is that it generally does not perform well on class-imbalanced downstream tasks. This is due to during supervised learning, the features are learned to tailor the class distribution from the pre-training set. It would be great that the authors can shed more light on this for any future improvement of the manuscript.

[A1] Lin et al. Feature Pyramid Networks for Object Detection. CVPR’17
[A2] Lee et al. Deeply-Supervised Nets. AISTATS’15


**Summary Of The Paper:**

The authors revisit the paradigm of supervised learning with recent successful breakthroughs in self-supervised learning, including multi-crop data augmentation [Caron et al., 2020], expendable projector head [Chen et al., 2020a], and a variant of nearest class means classifier [Mensink et al., 2012]. They found that those components, while working well with self-supervised settings, also work well with conventional supervised settings. Combined with those 3 components, the authors show improvement over other supervised, self-supervised, and semi-supervised approaches for ResNet-50.

**Summary Of The Review:**

Please provide a short summary justifying your recommendation of the paper.
The proposed modifications are simple and easy to understand. 3 components used in this work are well known from self-supervised learning and other fields. I do not see major novelty in the current form of the manuscript and encourage the authors to expand more on the class-imbalanced setting to see if the proposed changes can benefit for that scenario.

---

> ### Author Response · Authors · 2022-11-16
> **Author's Response**
>
> >  **Directly applying and combining 3 well known components in my opinion does not necessarily guarantee meeting the novelty bar at ICLR.**
>
> We respectfully disagree that the proposed approach and the extensive study presented in this paper “do not meet the novelty bar for ICLR”.
>
> a) The way we use projectors is by no means a “direct application” of previous work. We tested and reported ablations for a number of variants, with different depth, width, and normalization layers at different places. It is precisely because of these thorough ablations that we could uncover the crucial and novel insight that one can control the trade-off between training and transfer performance via the projector.
>
> b) The proposed OCM, a scalable online variant of NCM [Mensink et al., ECCV’12], which benefits from recent advances in SSL, is to the best of our knowledge novel. It successfully builds on the recent advances in SSL (e.g., momentum encoders) but its design is far from being a trivial update of NCM. Besides, the best models on both the training and transfer performance axes are OCM variants and as we see from Fig. 6-right, the learning dynamics of OCM and the standard case are different.
>
> c) Finally, beyond the contributions mentioned above, we believe that our empirical findings bring multiple key observations (summarized in Sec. 4.3) that are novel and valuable to the community. As a reminder, the main ones are:
>    - One should use labels, if available, for learning transferable representations, therefore challenging the recent finding that self-supervised learning approaches are better suited to pretrain for transfer tasks, and
>
>    - One can fully control the trade-off between training and transfer performance via the design of the expendable projector.
>
> > **It is known that aggressive cropping leads to better generalization, no surprise multi-crop works well for supervised learning.**
>
> We agree that it is not surprising that multi-crop helps supervised learning as well. However, this is not common practice in recent supervised learning papers like [LOOK, SL-MLP] and therefore we believe it is important to clearly showcase. Moreover, our paper is the first to study the effect of multi-crop hyperparameters _jointly_ with the use of an expendable projector (Fig. 4). It is also worth noting that, as we discuss in Sec. 4.1 and show in Table 8 (rows 4-5 versus row 8), aggressive cropping alone does not lead to strong gains; we see such gains when using multiple local crops and process them _at different resolutions_.
>
>
> > **The idea of using representations from earlier layers for downstream tasks has been largely explored  before e.g. in [A1, A2]**
>
> Indeed! As we discuss in the related work, several early self-supervised learning approaches  [Zhang et al., 2016; Gidaris et al., 2018] evaluate representations from multiple layers on image classification. Similarly, [A1] uses representations from earlier layers, but focuses on a different task (object detection). It also fine-tunes models before transfer and does not care about the pretraining task. Conversely, [A2] applies a margin-based supervised loss to multiple intermediate layer features, but focuses solely on the performance on the training datasets. It does not explore transfer learning.
>
> We thank the reviewer for suggesting these additional relevant references. We have updated our related work section to include them. Yet, none of their contributions overlap with ours.
>
> > **One can essentially use any combination of any previous layers for the downstream task training**
>
> This is true — layer combinations could be useful. However, finding the optimal combination of layers is far from trivial given the combinatorial nature of the problem, and this combination is likely to be heavily task dependent. As we show, adding an expendable projector instead is straightforward and performs very well both on the training and the transfer tasks.
>
> > **Supervised learning generally transfers poorly to class-imbalanced downstream tasks. Add evaluations on class-imbalanced downstream tasks.**
>
> Thanks for this comment and great suggestion.
> We have performed additional transfer experiments for **iNaturalist 2018** and **iNaturalist 2019**, two datasets that are commonly used as class-imbalanced downstream tasks, e.g., in RSB, DINO.
>
> We evaluated 5 models, DINO, PAWS, RSB-A1, t-ReX and t-ReX*, on these challenging datasets according to our linear logistic regression protocol. We report top-1 accuracy for each model in the table below:
>
> | Model | iNaturalist 2018 | iNaturalist 2019 |
> |---|:---:|:---:|
> | RSB-A1 | 34.9 | 43.2 |
> | PAWS | 40.8 | 49.8 |
> | DINO | 41.9 | 51.4 |
> | t-ReX | **45.8** | **54.2** |
> | t-ReX* | 36.0 | 44.2 |
>
> We see that our t-ReX and t-ReX* models still outperform RSB-A1 and DINO respectively, despite a challenging long-tail class distribution.
>
> We have added these results in the updated version of the submission in Section C.8 and Table 11.

---

> > ### Comment · Reviewer_yEpr · 2022-11-16
> > **Thanks for the respones**
> >
> > Thanks authors for the responses. All 4 reviewers raised the concern that this is mostly an empirical & engineering paper. The findings here might not be optimal when applied to different datasets/tasks. For example, the features from the last 2nd layers work well on dataset/task A might not be optimal for dataset/task B. However, I'm happy to see the additional results on class-imbalanced downstream tasks. I also agree with other reviewers that these empirical findings might be of interest to some of the ICLR's audience. Therefore I raised my score to 5. That being said, I won't actively argue for a rejection if the ACs' decision is otherwise.

---

### Official Review · Reviewer_o6eH · 2022-10-23

**Confidence:** 4
**Correctness:** 4
**Technical Novelty And Significance:** 2
**Empirical Novelty And Significance:** 2
**Recommendation:** 6

**Clarity, Quality, Novelty And Reproducibility:**

The overall analysis and paper structure are quite clear. The novelty is limited mainly due to the state of this paper being an analysis of existing training strategies rather than proposing a new framework. I have no questions on the reproduciblity.

**Strength And Weaknesses:**

For the full transparency, I reviewed the prior version of this work. The paper was rejected was mainly because of i) lack of insights of the motivation of each design and ii) lack the detailed analysis of each method. I am glad the current version has mostly resolved these issues and expanded the experiments significantly with more analysis and visualisations.

Strength.

++ A comprehensive analysis of training tricks for supervised image classification, which may benefits the community.

++ A detailed hyper-parameter ablative study on each training strategy to understand its contribution to the overall improvement.

Limitation.

-- All proposed training strategy is not new, and therefore the technical contribution is rather limited.

--The comparison for DINO and PAWS might be a bit unfair, since they are unsupervised learning methods?


**Summary Of The Paper:**

This paper proposes a supervised learning framework, named T-Rex, which is a combination of several training strategies, including 1) multi-scale cropping for data augmentation 2) a better designed projection head 3) an auxiliary loss function based on protypical contrastive learning. With the combination of these training strategies, the proposed method achieved improved performance on transfer learning and supervised learning benchmark on image classification tasks. The paper further analyse each component of the proposed training strategy and its corresponding hyper-parameters, shines some insights of the effectiveness of the overall improvement.

**Summary Of The Review:**

The paper is improved compared to the last version. The paper is clear in terms of explaining and providing detailed experiments on each training trick. The technical contribution is limited.

---

> ### Author Response · Authors · 2022-11-16
> **Author's Response**
>
> > **Updates over the prior version of this work**
>
> Thanks for this comment. We are glad that the reviewer appreciated all the work we put in improving our manuscript, including clarifying the text, expanding the experiments, and adding a new careful analysis of all the components of our approach.
>
> > **Proposed training is not new, technical contribution is limited**
>
> We agree that multi-crop, expendable projectors, and cosine CE have all been used individually in the past, although never jointly. Moreover, the proposed OCM, a scalable online variant of NCM [Mensink et al., ECCV’12] which benefits from recent advances in SSL, is novel. This variant brings a consistent gain (see Fig. 7), and exhibits a peculiar behavior that is interesting to study (see Fig 6-right).
>
> Even beyond the technical novelty itself, our thorough experimental study (as praised by all the four reviewers) uncovers multiple findings (see summary in Sec. 4.3) that we believe are important to share with the community. As a reminder, the main ones are: a) One should use labels, if available, for learning transferable representations, therefore challenging the recent finding that self-supervised learning approaches are better suited to pretrain for transfer tasks, and b) one can fully control the trade-off between training and transfer performance via the projector.
>
> > **Comparison to DINO and PAWS is unfair, since they are unsupervised methods**
>
> If we understood correctly, the fairness mentioned here is related to the fact that our method uses labels while DINO and PAWS do not.
>
> We would like to stress that our paper studies performance on not only the training task but also transfer ones. For the latter, previous work [Sariyildiz et al., ICCV’21 among others] experimentally showed that self and semi-supervised methods are state-of-the-art for transfer, performing better than methods using labels. In this paper, we challenge this observation, and show that conclusions are highly tied to the model architecture, in particular to the projector.
>
> Although comparisons to supervised methods would be unfair if we were only looking at performance on ImageNet-1k, here we thoroughly explain how labels, which often lead to overfitting to the training task and therefore to worse generalization, can be properly used to improve on both fronts.

---

> > ### Comment · Reviewer_o6eH · 2022-11-26
> > **Response to the rebuttal**
> >
> > Thanks to the authors for the rebuttal. All my concerns have been resolved. And I would love to keep my weak acceptance as my final rating.

---

### Official Review · Reviewer_63Ba · 2022-10-24

**Confidence:** 3
**Correctness:** 4
**Technical Novelty And Significance:** 2
**Empirical Novelty And Significance:** 3
**Recommendation:** 8

**Clarity, Quality, Novelty And Reproducibility:**

The paper is clearly written, and I enjoyed reading it.

I would probably use u and w for Figure 2 instead of mu and omega, as they are consistent with the equations in the text.

The discussion about gradient similarity and the middle figure in Figure 6 were not very intuitive to me. Intuitively, I would expect the gradients to be similar for easy samples and to be diverse for challenging samples; since Base+Mc would be more difficult than Base, the high similarity of Base+Mc would seem to contradict this. The middle figure in Figure 6 may just show that Base converges faster. I would like to ask the authors why the gradient similarity is higher in Base+Mc than in Base.


**Strength And Weaknesses:**

The strengths of this paper are (1) finding that generalization can be obtained from multi-crops data augmentation (DA) that SSL is based on, (2) finding that deeper head leads to better generalization and shallower head leads to better IN1K performance, (3) presenting SOTA models (t-ReX and t-ReX*) for the training task vs transfer task performance, and (4) an interesting comparison between the use of class weights and the use of prototypes for training, showing that prototypes with a memory bank performs slightly better.

The first and second findings are supported by several careful analyses as well as direct experiments. They show that the intra-class L2 distance between samples increases with the projector and decreases with the multicrop DA, and that the fraction of feature dimensions close to zero (which they call sparsity) is larger without the projector. These results indicate that deeper projectors improve generalization performance because the representation is more distributed and less sparse. Those analyses were interesting to me. The SOTA models will be released, and I think the models as well as the details of the learning presented in this paper will be valuable to the community.

The weaknesses of this paper are that the methodology is one that the community is already familiar with and the findings are still based on experiments, so the findings have not been shown to be theoretically correct. We can make hypotheses, but we do not know if the findings are correct for other large data sets.


**Summary Of The Paper:**

Motivated both by the fact that Self-Supervised Learning (SSL) models have better transferability than supervised models and by the expectation that additional information on labels should not impair generalization performance, this paper attempts to achieve both good supervised classification accuracy and good transferability to other tasks. Regarding the trade-off between good performance on the training task and transferability, the paper carried out extensive experiments and claims that multi-crop data augmentation is one of the key ingredients for transferability. It also finds that the trade-off can be controlled by the design of the projector head. Deeper head leads to better generalization and shallower one leads to better IN1K performance. The paper also incorporates a class prototype to improve performance and achieves SOTA regarding training-versus-transfer performance.

**Summary Of The Review:**

Overall, the paper was an enjoyable read. Although only empirical findings are presented, the experiments are careful and reliable. The findings on transfer learning will be valuable to a wide range of audience.

---

> ### Author Response · Authors · 2022-11-16
> **Author's Response**
>
> > **The methodology is one that the community is already familiar with.**
>
> We agree that multi-crop, expendable projectors, and cosine CE have all been used individually in the past, although never jointly. Moreover, the proposed OCM, a scalable online variant of NCM [Mensink et al., ECCV’12] which benefits from recent advances in SSL, is novel. This variant brings a consistent gain (see Fig. 7), and exhibits a peculiar behavior that is interesting to study (see Fig 6-right).
>
> Even beyond the technical novelty itself, our thorough experimental study (as praised by all the four reviewers) uncovers multiple findings (see summary in Sec. 4.3) that we believe are important to share with the community.
> As a reminder, the main ones are:
>    a) One should use labels, if available, for learning transferable representations, therefore challenging the recent finding that self-supervised learning approaches are better suited to pretrain for transfer tasks, and
>    b) one can fully control the trade-off between training and transfer performance via the projector.
>
> > **The findings are based on experiments, no theory behind.**
>
> We fully agree on the experimental nature of our contributions. Yet, we believe there exists key insights in our empirical findings that are important to share with the community. Maybe more importantly, our extensive experimental results are fully supported by “several careful analyses” (as noted by the reviewer). This is part of our effort towards going deeper than merely showing performance gains, and getting a better understanding of the underlying mechanisms.
>
> Although highly desirable, developing a theoretical understanding of our components (e.g., non-linear projectors) is very challenging. Some recent works attempt to explain e.g., the reason why non-linearity in projectors avoids dimensional collapse [Jing et al, 2022], we note that the theory works only for the linear case, and the authors still rely on interesting empirical observations to derive their method.
>
> Jing et al. (2022)  "Understanding dimensional collapse in contrastive self-supervised learning." ICLR 2022
>
> > **We do not know if the findings are correct for other large data sets.**
>
> This is indeed an open question.
>
> We performed our analysis on ImageNet-1K as this is the de facto dataset for visual representation learning. This allows us to fairly and directly compare to all recent contributions to this field. Duplicating the full study for another large-scale dataset would be infeasible in practice. Comparisons to DINO, PAWS, SupCon, LOOK, RSB would require a prohibitive amount of compute to not only train but also properly set hyper-parameters (note that when using public models, we are guaranteed to start from the strongest version of each method we compare to).
>
> Yet, to partially address this comment, we trained our t-ReX model on the full ImageNet dataset (using the images for the 19K concepts available in [the last release of ImageNet](https://image-net.org/update-mar-11-2021.php)).
>
> The table below reports preliminary results, following the evaluation protocol introduced in the paper, for two models: t-ReX as already defined in the paper (trained on ImageNet-1K), and a new t-ReX flavor trained on the full ImageNet dataset (ImageNet-19K). Note that, since the ImageNet-CoG datasets are now part of the training set, we exclude them when computing transfer performance.
>
> | Model | Training Dataset | ImageNet-1K (Top-1 Accuracy) | Average Transfer (Log odds, excluding CoG datasets) |
> |---|---|:---:|:---:|
> | t-ReX | ImageNet-1K | 78.0 | 1.8056 |
> | t-ReX | ImageNet-19K | 78.5 | 2.1506 |
>
> We observe stronger transfer learning performance when training on a superset of ImageNet-1K, i.e. a much larger dataset.
>
>
>
> > **Use u and w for Figure-2**
>
> Thanks for noticing this inconsistency in Fig 2, we fixed it in the updated manuscript.
>
> >  **Why the gradient similarity is higher in Base+Mc than in Base**
>
> Thanks for this great question.
>
> In Fig 6 (middle), we plot the average cosine similarity between the gradients of all pairs of class weights at each iteration.
> We plot the average similarity for two models, one with multi-crop (Base+Mc) and one without (Base).
>
> This figure conveys that the average cosine similarity between gradients of individual class weights increases substantially with multi-crop (Base+Mc). In other words, on average, classifier gradients (and therefore the class weights themselves) are more entangled. We attribute this to the fact that some of the local crops (e.g., the ones that mostly cover background and hence are not really discriminative for the class at hand) are harder to classify. This leads to a harder task and gradients of higher variance (shown also in Fig 16).
>
> We have clarified this in the updated manuscript.

---

> > ### Comment · Reviewer_63Ba · 2022-11-21
> > **Thank you for the response**
> >
> > They clearly answered my questions; additional experiments with IN19K showed improved transferability, which more concretely supported their claims. The explanation to my question about the gradient similarity addressed my concern. I would like to recommend acceptance of the paper to ICLR.

---

### Author Response · Authors · 2022-11-16
**Author's Comment After Reviews**

First of all, we would like to warmly thank all reviewers for thorough reviews and constructive feedback.

We are thrilled that overall the feedback is positive.

All the reviewers praised our experimental analysis, finding it “extensive”. R-63Ba mentions that “findings are supported by several careful analyses”, arguing that the findings will be “useful to the community” (also mentioned by R-o6eH) and “to a wide range of audience”. R-o6eH mentions that our analysis “shines some insights of the effectiveness” of our approach. R-yEpr and R-o6eH praised our ablation study.  R-p7Yg mentions that our “empirical evaluation is extensive and does adequately support the claims” and leads to “SOTA models on each of the two settings”.

Finally, three of the four reviewers mention that the paper is clearly written (R-63Ba, R-o6eH, R-p7Yg).

We have responded to each reviewer individually below. We hope that these responses clarify any remaining questions. If needed, we would be happy to engage in further discussion to clarify any point that would still be unclear.

We have also uploaded an updated version of our manuscript. The main changes (new text colored in blue for convenience) are that:
- We clarified the text referring to the gradient similarity plot, following R-63Ba’s comment
- We added references [A1], [A2] as suggested by R-yEpr
- We added transfer learning results for two class-imbalanced downstream tasks: iNaturalist-2018 and iNaturalist-2019 (Table 11), as suggested by R-yEpr

---

### Decision · Program_Chairs · 2023-01-20

**Decision:**

Accept: notable-top-25%

**Justification For Why Not Higher Score:**

The results are interesting, but not groundbreaking.

**Justification For Why Not Lower Score:**

This is a nice methodologically clean study and could make for a good spotlight.

**Metareview: Summary, Strengths And Weaknesses:**

The paper explores applying tricks from self-supervised learning literature (multi-crop, disposable projection head) to supervised learning on ImageNet1k and explores the tradeoff between the training task performance and the transfer performance. This results in a "frontier" of models trading off the two performance values.

The reviewers are mostly positive about the paper, with only one of them leaning toward rejection. The main arguments are as follows:

Pros:
- An interesting and potentially practically relevant problem setting
- Clean and extensive experiments
- Good presentation

Cons:
- The components used in the models are not really new
- Not clear to which degree the results would apply to training on other models and datasets

All in all, I recommend acceptance. The paper suggests decoupling the training objective (self-supervised vs supervised) from extra tricks (multi-crop, projection head) and via comprehensive experiments shows that oftentimes it's the tricks that matter. This investigation adds to our understanding of pre-training and can be of interest to the ICLR audience. Moreover, the rebuttal addressed the reviewers' concerns fairly convincingly.

**Note From Pc:**

if the above contains the word "oral" or "spotlight" please see: "oral" presentation means -> notable-top-5% and "spotlight" means -> notable-top-25%. As stated in our emails, we are disassociating presentation type from AC recommendations